# The LINC complex ensures accurate centrosome positioning during prophase

Joana T Lima[1,2,3], António J Pereira[1], Jorge G Ferreira[1,2]

**Accurate centrosome separation and positioning during early mitosis relies on force-generating mechanisms regulated by a combination of extracellular, cytoplasmic, and nuclear cues. The identity of the nuclear cues involved in this process remains largely unknown. Here, we investigate how the prophase nucleus contributes to centrosome positioning during the initial stages of mitosis, using a combination of cell micropatterning, high-resolution live-cell imaging, and quantitative 3D cellular reconstruction. We show that in untransformed RPE-1 cells, centrosome positioning is regulated by a nuclear signal, independently of external cues. This nuclear mechanism relies on the linker of nucleoskeleton and cytoskeleton complex that controls the timely loading of dynein on the nuclear envelope (NE), providing spatial cues for robust centrosome positioning on the shortest nuclear axis, before nuclear envelope permeabilization. Our results demonstrate how nuclear–cytoskeletal coupling maintains a robust centrosome positioning mechanism to ensure efficient mitotic spindle assembly.**

## Introduction

Mitosis is the process by which a cell divides its genetic content into two identical daughter cells. The start of this process is typically defined as the moment when cells start to condense their chromosomes inside the nucleus (Maddox et al, 2006; Antonin & Neumann, 2016). This occurs with a near-simultaneous remodelling of their cytoskeleton and a decrease in the cell–matrix adhesion (Dao et al, 2009), a process regulated via integrin and cadherin signalling (Mui et al, 2016). Cytoskeletal restructuring is a consequence of the dynamic changes that occur in the microtubule (Zhai et al, 1996; Mchedlishvili et al, 2018) and actomyosin networks (Maddox & Burridge, 2003; Chugh & Paluch, 2018), required to build a mitotic spindle and a stiff mitotic cortex. Subsequently, nuclear pore complexes (NPCs) (Linder et al, 2017) disassemble and the nuclear lamina depolymerizes (Heald & McKeon, 1990), resulting in nuclear envelope permeabilization (NEP). These global changes are controlled by mitotic kinases such as PLK1 and CDK1 (Gavet & Pines, 2010; Gheghiani et al, 2017; Jones et al, 2018), the latter ensuring coordination of cytoplasmic and nuclear events (Gavet & Pines, 2010).

The goal of mitosis is to ensure the accurate segregation of chromosomes. This requires the assembly of a microtubule-based mitotic spindle that interacts with chromosomes via the kinetochores (Toso et al, 2009; Thompson & Compton, 2011). In human cells, the establishment of a bipolar spindle relies primarily on centrosomes (Tanenbaum & Medema, 2010; Agircan et al, 2014) that separate along the nuclear envelope (NE) during prophase, through forces generated by motor proteins, such as kinesin-5 and dynein (Raaijmakers et al, 2012; van Heesbeen et al, 2013). Importantly, the extent of centrosome separation and their positioning at the moment of NEP has been shown to influence mitotic fidelity (Whitehead et al, 1996; Kaseda et al, 2012; Silkworth et al, 2012; Nunes et al, 2020) and mitotic timing (Nunes et al, 2020).

Once a bipolar spindle is assembled, it must then orient inside the cell, to define a division axis. The involvement of cortical cues in mitotic spindle orientation during metaphase is well established (Théry et al, 2005, 2007; Petridou & Skourides, 2014). This mechanism requires the localization of the LGN-Gα1-NuMA complex to the cell cortex, which then recruits dynein to generate pulling forces on astral microtubules. However, during the early stages of mitosis this process appears to be fundamentally different. Accordingly, we have shown that during prophase, NE-associated dynein, together with Arp2/3 activity, is required to position centrosomes in the shortest nuclear axis, independently of cortical dynein (Nunes et al, 2020). These observations suggest that a nuclear cue is required for accurate centrosome positioning before NEP. However, the molecular nature of this nuclear cue is still unclear.

The nucleus and some of its components have already been proposed to impact centrosome function. Accordingly, perturbations in the nuclear lamina because of the loss of lamin A lead to impaired centrosome separation and asymmetric NPC distribution in prophase (Guo & Zheng, 2015; Boudreau et al, 2019). In addition, different components of the linker of nucleoskeleton and cytoskeleton (LINC) complex, required for nucleocytoplasmic coupling (Lombardi et al, 2011), have been implicated in centrosome–nucleus

[1]Instituto de Investigação e Inovação em Saúde (i3S), Porto, Portugal   [2]Departamento de Biomedicina, Faculdade de Medicina do Porto, Unidade de Biologia Experimental, Porto, Portugal   [3]Programa Doutoral em Biomedicina, Faculdade de Medicina, Universidade do Porto, Porto, Portugal

Correspondence: jferreir@ibmc.up.pt

tethering during cell migration, by directly associating with NE-bound dynein (Malone et al, 2003; Zhang et al, 2009; Splinter et al, 2010; Boudreau et al, 2019). Moreover, the LINC complex also facilitates centrosome separation during prophase (Stiff et al, 2020) and decreases chromosome scattering during prometaphase, aiding their capture after NEP (Booth et al, 2019). Nevertheless, how these cytoplasmic and nuclear players interact during the G2/M transition to ensure accurate centrosome positioning and efficient spindle assembly remains unknown. Here, we identify the LINC complex as essential for defining the centrosome–nucleus axis during prophase in normal, near-diploid cells, by orienting centrosomes towards the shortest nuclear axis.

# Results

## Chromosomally stable RPE-1 cells systematically position their centrosomes at the shortest nuclear axis

It has been previously shown that centrosome separation and positioning at the moment of NEP impacts mitotic progression and fidelity (Kaseda et al, 2012; Silkworth et al, 2012; Nunes et al, 2020). However, the degree of conservation of this mechanism between normal, untransformed cells and cancer cell lines is still unknown. To address this, we performed live-cell imaging of a chromosomally stable RPE-1 cell line and two chromosomally unstable cancer cell lines of different origins (U2-OS derived from osteosarcoma and MDA-MB-468 derived from breast adenocarcinoma), with high spatial and temporal resolution. All cell lines were seeded on 10-$\mu$m-wide micropatterned lines coated with fibronectin (FBN; Fig 1A). This allowed us to accurately standardize cell shape and intracellular organization (Théry et al, 2005; Nunes et al, 2020). Then, we reconstructed the dynamics of centrosomes, nucleus, and cell membrane during the G2-M transition, using our previously developed tool, Trackosome (Castro et al, 2020) (Fig S1A–E). With this approach, we can obtain quantitative data and correlate several parameters; namely, we determined the alignment of the longest cell axis with the longest nuclear axis (Fig S1F), correlated the alignment of a vector connecting the two centrosomes with the longest nuclear axis (angle centrosome–nucleus, Fig S1G), and measured the separation of the two centrosomes to opposite sides of the nucleus, by determining the angle formed by a vector that connects the two centrosomes that intercepts the nuclear centroid (angle centrosome–centrosome, Fig S1H). This allowed us to construct polar plots that display the alignment of the centrosomes with the longest nuclear axis (Fig S1I; note that an orientation towards 90° corresponds to alignment with the shortest nuclear axis) and graphs that correlate centrosome positioning and separation to opposite sides of the nucleus (Fig S1J).

A detailed analysis of RPE-1 cells during the G2-M transition revealed that the long axis of the nucleus is initially aligned with the long cell axis (Fig 1A, left panel, and Fig 1B; Video 1), likely because of a geometrical constraint imposed by the line micropattern, as was previously shown for interphase cells (Versaevel et al, 2012). Once cells start to round up in preparation for mitosis, as measured by the decrease in cell membrane eccentricity, the long nuclear axis

progressively diverges from the long cell axis (Fig 1B), suggesting that mitotic cell rounding uncouples the two parameters. This reorientation occurs simultaneously with the movement of centrosomes to opposite sides of the nucleus (Fig 1C). Consequently, at NEP, centrosomes are positioned on the shortest nuclear axis (Fig 1C and D). Importantly, this behaviour is not observed in the two cancer cell lines analysed. In U2-OS cells, mitotic cell rounding is altered in comparison with RPE-1 cells (Fig 1A, middle panel; Fig 1B and E; ****P < 0.001) and the nucleus maintains alignment with the main cell axis almost until the moment of NEP (Fig 1E; Video 2; ****P < 0.001). As a result of this physical constraint, centrosomes separate, but fail to position on the shortest nuclear axis at NEP (Fig 1F and G; *P = 0.022). On the contrary, MDA-MB-468 cells are morphologically distinct from RPE-1 or U2-OS. They are rounder and have a smaller area of adhesion to the micropatterned substrate (Fig 1A, right panel; Video 3). As a result, they do not exhibit the decrease in cell membrane eccentricity normally observed during mitotic cell rounding (Fig 1B and H; ****P < 0.001). Consequently, the long nuclear axis is randomly oriented inside the cell (Fig 1H; P < 0.05). In addition, centrosomes in MDA-MB-468 often show incomplete separation and are unable to reach opposite sides of the nucleus at NEP (Fig 1I). This defect in centrosome separation could be due to the decrease in cell adhesion, which affects the activity of kinesin-5 (Nunes et al, 2020; Kamranvar et al, 2022), essential for the initial stages of centrosome separation. Consequently, centrosomes are randomly positioned relative to the nucleus at the moment of NEP (Fig 1J; **P = 0.0028). Overall, our data show that RPE-1 cells have a robust centrosome positioning mechanism during the G2-M transition, and this mechanism is compromised both in U2-OS and in MDA-MB-468 cells.

## Mitotic cell rounding is not the major determinant for centrosome positioning

Given that the U2-OS and MDA-MB-468 cell lines that we tested failed to correctly position centrosomes on the shortest nuclear axis and displayed altered mitotic cell rounding, we wondered whether the two events were interconnected. This is particularly relevant because it was previously shown that mitotic rounding is essential for providing the necessary space to assemble a mitotic spindle (Lancaster et al, 2013; Sorce et al, 2015; Dix et al, 2018; Nunes et al, 2020). To test this hypothesis, we proceeded to interfere with mitotic cell rounding in RPE-1 cells, by either impairing it or accelerating it. Mitotic rounding requires both cortical retraction and adhesion disassembly. Therefore, we started by acutely treating cells with a Rho-associated protein kinase inhibitor (Y-27632; Fig 2A and B), known to decrease actomyosin contractility and delay cortical retraction (Maddox & Burridge, 2003). Upon treatment with Y27632, centrosomes were still able to separate and position correctly on the shortest nuclear axis (Fig 2C–F), although the rate of cell rounding was delayed because of an impairment of cortical retraction (Fig 2G; ***P < 0.001). This is in agreement with our previous observations in HeLa cells (Nunes et al, 2020). Next, we proceeded to interfere with adhesion disassembly by expressing a mutant form of Rap1 (Rap1Q63E; Rap1*), which effectively blocks mitotic rounding (Dao et al, 2009) (Fig 2H–J; ***P < 0.001). Delaying adhesion disassembly in RPE-1 cells did not affect centrosome

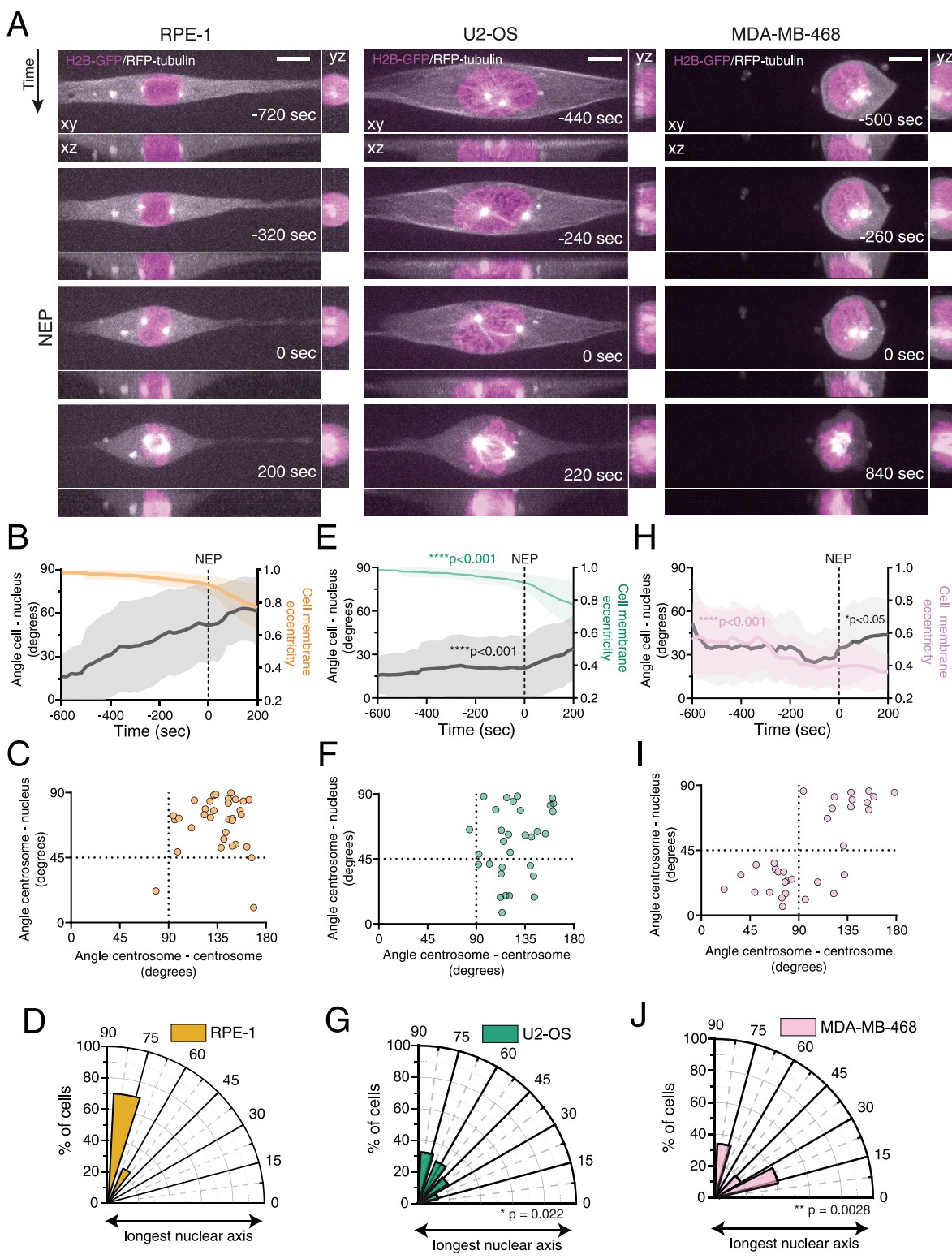

**Figure 1. Chromosomally stable, near-diploid cells efficiently separate and position their centrosomes at the shortest nuclear axis.**
**(A)** Representative frames from a movie of a near-diploid, untransformed RPE-1 cell (left panel) and chromosomally unstable U2-OS (middle panel) and MDA-MB-486 (right panel) cancer cells, seeded on a 10-μm-wide micropatterned line, showing centrosome movement and the overall cellular reorganization in preparation for mitotic entry. Time is in sec, and time zero corresponds to nuclear envelope permeabilization (NEP). Scale bar = 10 μm. **(B)** Quantification of cell membrane eccentricity (orange) and the angle between the longest cell axis and longest nuclear axis (angle cell–nucleus; grey), of RPE-1 cells during mitotic entry, obtained with our custom-made MATLAB script. The line represents the mean value, and the shaded area corresponds to SD (n = 33). The dashed line represents the moment of NEP. **(C)** Correlation between centrosome separation to opposite sides of the nucleus (angle centrosome–centrosome; x-axis) and centrosome

separation or positioning on the shortest nuclear axis (Fig 2K–N; $P$ = 0.583), indicating that cell rounding impairment in RPE-1 cells does not significantly impact centrosome positioning at the moment of NEP.

Next, we tested the effect of inducing premature rounding by treating cells with calyculin-A (CalA), a protein phosphatase inhibitor that increases the phosphorylation of ezrin/radixin/moesin (ERM) proteins and triggers cell rounding in adherent cells (Tachibana et al, 2015). Acute treatment with CalA led to a faster rounding of the cells (Fig 2O and P; \*\*\*$P$ < 0.001), as anticipated. Nevertheless, this did not significantly affect centrosome separation or positioning at NEP (Fig 2Q and R; $P$ = 0.162). Overall, we conclude that centrosome positioning in RPE-1 cells during early mitosis is a robust process, which is largely independent of mitotic cell rounding.

To further investigate the relevance of cell rounding for centrosome positioning, we then decided to interfere with mitotic cell rounding in the U2-OS and MDA-MB-468 cancer cell lines. We either forced adhesion in the highly rounded MDA-MB-468 or promoted rounding in the flatter U2-OS cells. We started by transfecting MDA-MB-468 cells with Rap1\*. As expected, Rap1\* expression delayed cell rounding (Fig S2A and B; \*\*\*\*$P$ < 0.001), but this did not restore correct centrosome separation or positioning (Fig S2C and D). Next, we tried to interfere with rounding in U2-OS cells using Y-27632 or CalA (Fig S2E–I). Neither treatment was sufficient to restore centrosome positioning on the shortest nuclear axis (Fig S2J–M). Because these results were obtained from cells seeded on 10-$\mu$m-wide micropatterned lines, we also wanted to rule out the possibility that the failure to position centrosomes could be due to geometrical constraints imposed by the micropattern. Therefore, we seeded U2-OS cells on 20-$\mu$m-wide micropatterned lines or on non-patterned FBN-coated coverslips. In all these conditions, U2-OS cells were still unable to correctly position their centrosomes at the shortest nuclear axis (Fig S2N–S), even though they showed a significant decrease in cell membrane eccentricity when placed in non-patterned substrates, reflecting a more rounded state (Fig S2T; \*\*\*\*$P$ < 0.0001). Taken together, our results indicate that centrosome positioning on the shortest nuclear axis in normal, untransformed RPE-1 cells is independent of the rounding state of the cell and that the defects in positioning observed in the U2-OS and MDA-MB-468 cancer cells cannot be rescued by manipulating cell rounding.

### The nuclear lamina is not required for centrosome positioning on the shortest nuclear axis

Given these results and our previous observations (Nunes et al, 2020), it seems unlikely that an external cue could be driving centrosome positioning during early mitosis. Therefore, we hypothesized that a nuclear cue could be responsible for ensuring accurate centrosome positioning on the shortest nuclear axis. One possible candidate is the nuclear lamina, because it was previously shown that lamins can position NPCs and centrosomes (Guo & Zheng, 2015) and are required for centrosome separation (Boudreau et al, 2019). We started by analysing the levels of nuclear lamina components in each of the cell lines in our panel, by measuring the fluorescence intensity of lamin A/C and lamin B1 at the NE (Fig S3A and B). We observed that the levels of both lamin A/C and lamin B1 were altered in U2-OS and MDA-MB-468, when compared to RPE-1 cells (Fig S3C and D; \*\*\*\*$P$ < 0.001). Consequently, the ratios of lamin A/C relative to lamin B1 in the two cancer cell lines were substantially lower than in RPE-1 cells (Fig S3E). Our results are in agreement with lamin expression data available from the online repository Cancer Dependency Map Project (DepMap) (Fig S3F). Moreover, the decrease in lamin A levels observed in cancer cells was accompanied by a decrease in nuclear solidity (Fig S3G; \*\*\*\*$P$ < 0.0001), which suggests that nuclear structure in both U2-OS and MDA-MB-468 cells is altered when compared to RPE-1 cells. Thus, we decided to manipulate lamin levels in RPE-1 cells and ask whether this was sufficient to impair centrosome positioning on the shortest nuclear axis. We started by depleting lamin A in RPE-1 cells, using RNA interference (siLMNA; Fig 3A–C). However, this depletion was not sufficient to alter centrosome behaviour, as cells still efficiently separated and positioned them at the shortest nuclear axis (Fig 3D–G). To confirm our observations, we next sought to imbalance the lamin A:B ratio by overexpressing a HaloTag9-lamin B1 construct in RPE-1 cells (Figs 3H and I and S3H; \*\*\*\*$P$ < 0.0001). Similar to lamin A depletion, lamin B1 overexpression also did not disrupt correct centrosome positioning at NEP (Fig 3J–M). Taken together, these results indicate that even though the lamin A:B ratio is perturbed in the two cancer cells used for this study, this is likely not the cause for the observed errors in centrosome positioning at NEP, as experimental manipulation of the levels of either lamin A or B in RPE-1 cells did not affect this process. Next, we sought to interfere with additional nuclear components, independently of lamin levels. To do so, we overexpressed a mCherry-tagged version of the lamin B receptor (LBR), an integral protein of the inner nuclear membrane that associates with the nuclear lamina (Fig S3I; \*\*\*\*$P$ < 0.0001) (Appelbaum et al, 1990). It has been previously shown that LBR overexpression causes perinuclear ER expansion and the overproduction of nuclear membranes (Ma et al, 2007), leading to NE folding and altered nuclear structure (Dantas et al, 2022), similar to what we observed in U2-OS and MDA-MB-468 cells. However, interfering with the nuclear structure by overexpressing LBR in RPE-1 cells (Fig 3N and O) did not affect the separation and positioning of centrosomes on the shortest nuclear

positioning relative to the longest nuclear axis (angle centrosome–nucleus; y-axis), at the moment of NEP for RPE-1 cells. Cells that efficiently separate their centrosomes will present high values of centrosome–centrosome angle, and therefore are located at the right half of the graph. Cells that correctly position their centrosomes at the shortest nuclear axis will cluster at the upper half of the graph. **(D)** Polar plot quantifying centrosome positioning relative to the longest nuclear axis at NEP for RPE-1 cells. **(E)** Quantification of cell membrane eccentricity (green; \*\*\*\*$P$ < 0.001) and angle cell–nucleus (grey; \*\*\*\*$P$ < 0.0001) for U2-OS cells (n = 31). **(F)** Correlation between the angle centrosome–centrosome (\*$P$ = 0.036) and the angle centrosome–nucleus for U2-OS cells. **(G)** Polar plot quantifying centrosome positioning relative to the longest nuclear axis at NEP for U2-OS cells (\*$P$ = 0.022). **(H)** Quantifications of cell membrane eccentricity (pink; \*\*\*\*$P$ < 0.0001) and angle cell–nucleus (grey; \*$P$ = 0.011) for MDA-MB-468 cells (n = 32). **(I)** Correlation between the angle centrosome–centrosome (\*\*\*\*$P$ < 0.001) and the angle centrosome–nucleus, for MDA-MB-468 cells. **(J)** Polar plot quantifying centrosome positioning relative to the longest nuclear axis at NEP for these MDA-MB-468 cells (\*\*$P$ = 0.028).

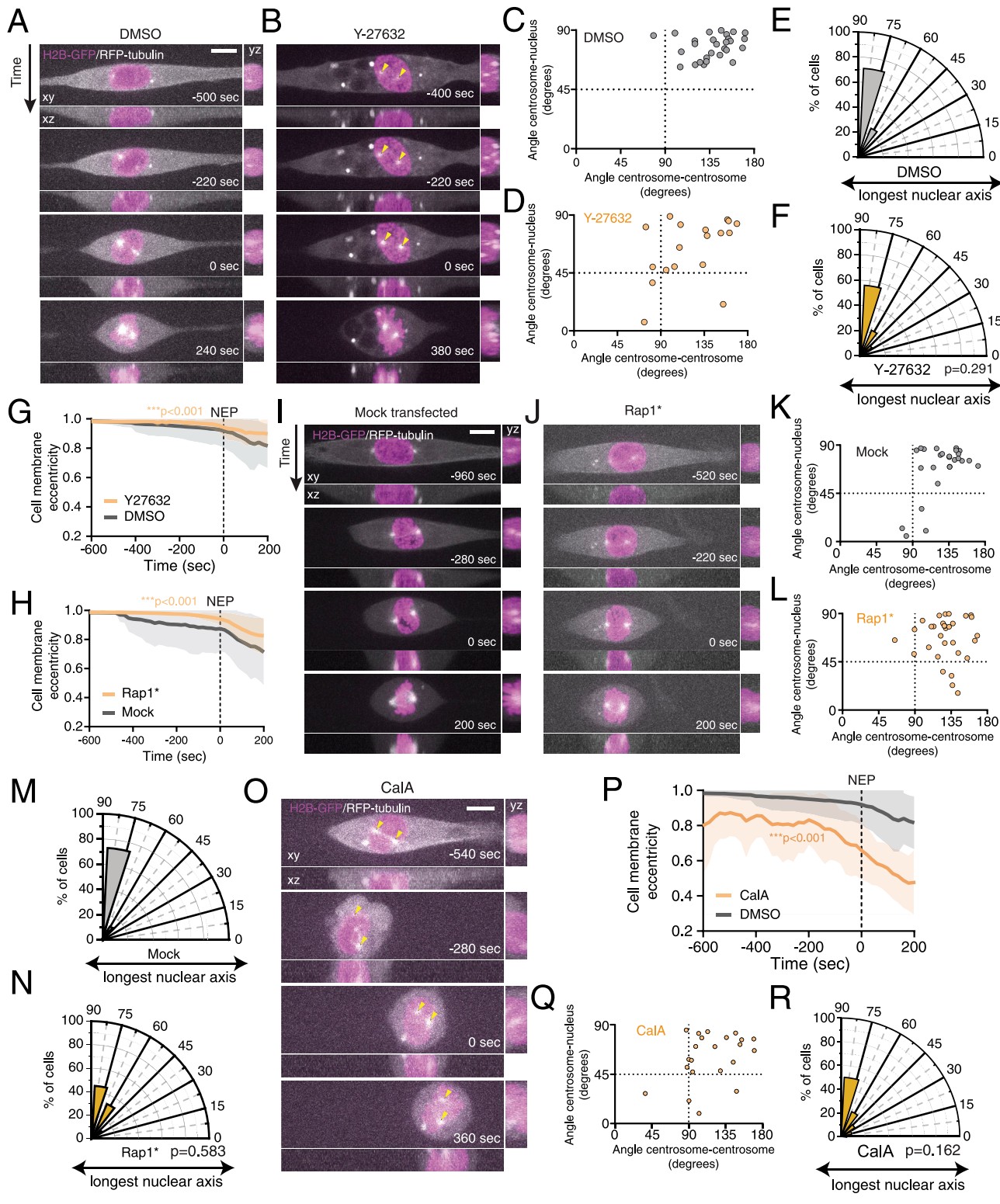

**Figure 2. Cell rounding status does not influence centrosome positioning in RPE-1 cells.**
**(A, B)** Representative frames from a movie of an RPE-1 cell stably expressing H2B-GFP and RFP-tubulin, seeded on a 10-$\mu$m-wide micropatterned line, treated with (A) DMSO and (B) Y-27632, showing centrosome movement. Yellow arrowheads indicate centrosome position. **(C, D)** Correlation between centrosome separation (angle centrosome–centrosome; x-axis) and centrosome positioning (angle centrosome–nucleus; y-axis), at the moment of nuclear envelope permeabilization (NEP), for DMSO-treated ((C); n = 28) or Y-27632-treated RPE-1 cells ((D); n = 18). **(E, F)** Polar plot quantifying centrosome positioning relative to the longest nuclear axis at NEP for these RPE-1 cells treated with DMSO (E) or Y-27632 ((F); P = 0.291). **(G, H)** Quantification of cell membrane eccentricity of DMSO-treated cells ((G); grey) and Y-27632-treated cells (orange; ****P < 0.001); or mock-transfected ((H); mock; grey; n = 25) and Rap1*-transfected (orange; n = 31; ****P < 0.001) cells. **(I, J)** Representative frames from movies of

axis (Fig 3P–S). Overall, our data indicate that neither the nuclear lamina nor LBR is required for correct centrosome positioning during early mitosis. They further suggest that the altered lamin levels in U2-OS and MDA-MB-468 are likely not the cause for the mispositioned centrosomes observed in these cell lines.

### The LINC complex is required for centrosome positioning on the shortest nuclear axis

Next, we focused our attention on the LINC complex, because it was previously shown to play a role in early mitosis (Booth et al, 2019; Stiff et al, 2020). Immunolocalization of SUN1 and SUN2 in prophase cells revealed that both proteins localized to the NE, as expected (Fig 4A and B). Interestingly, in a significant proportion of cells, SUN1 and SUN2 also concentrated in the area between and around the two separating centrosomes and their corresponding microtubule arrays (Fig 4A and B, insets). This localization was dependent on microtubules, because treatment with low doses of nocodazole (Noc) disrupted inter-centrosomal SUN localization (Fig 4C and D). We then proceeded by depleting individual SUN proteins (SUN1 and SUN2) in RPE-1 cells using a lentiviral-mediated shRNA, giving rise to a heterogeneous population of cells with different levels of depletion (Figs 4A and B and S4A–D). This was sufficient to decrease NE localization and inter-centrosomal accumulation of SUN1 and SUN2 (Fig 4A and B). We then assessed the impact of SUN1 and SUN2 depletion on centrosome behaviour throughout mitotic entry. Upon depletion, these cells were still capable of efficiently separating centrosomes (Fig 4E–J; Video 4 and Video 5). However, they had an impaired positioning of centrosomes at the shortest nuclear axis (Fig 4K–M; *P = 0.0165 and P = 0.0175 for SUN1 and SUN2, respectively).

Given the results obtained with SUN1 and SUN2, we then tested whether nesprins, which are part of the LINC complex and interact with SUN proteins in the perinuclear space, could also impact centrosome function. For that purpose, we expressed a dominant-negative form of KASH tagged with mCherry (DN-KASH; Fig S4E; Video 6 and Video 7) that prevents the binding of endogenous nesprins to SUN proteins and blocks force propagation across the NE (Lombardi & Lammerding, 2011; Zhang et al, 2019). As a corresponding control, we generated a cell line expressing a mCherry-tagged mutant version of the same DN-KASH construct where the last four amino acids (PPPL) of the KASH domain were removed (ΔPPPL-KASH). This prevents ΔPPPL-KASH from interacting with SUN1 or SUN2, thus making it a "defective" dominant-negative (Zhang et al, 2019). Firstly, we confirmed by immunofluorescence the ability of the DN-KASH construct to displace endogenous nesprins from the NE, as assessed by nesprin-2 localization (Fig 5A; SYNE2). Contrarily, and as expected, the expression of the ΔPPPL-KASH construct did not interfere with nesprin localization on the NE, similar to unmanipulated RPE-1 cells

(Figs 5A and S4F). We then analysed centrosome behaviour in these cells. Upon the expression of ΔPPPL-KASH, cells behaved similar to unmanipulated RPE-1 cells, with an efficient separation and positioning of centrosomes on the shortest nuclear axis (Figs 1C and 5B–D; P = 0.708 for ΔPPPL-KASH compared with unmanipulated RPE-1 cells). However, DN-KASH–expressing cells showed compromised separation and positioning of centrosomes (Fig 5E–G; *P = 0.0155 and *P = 0.0237, respectively).

Finally, we decided to test whether the cancer cell lines used in this study have an altered LINC complex, considering that they show centrosome positioning defects (Fig S5). To achieve this, we quantified the levels of different LINC complex components on the NE by performing immunofluorescence to detect SUN1, SUN2, SYNE1 (nesprin-1), and SYNE2 (nesprin-2). Our results indicate that U2-OS cells have a significant decrease in the levels of nesprin-1 and SUN2 (Fig S5A, B, and E; ****P < 0.0001), whereas MDA-MB-468 show a decrease in nesprin-1, SUN1, and SUN2 (Fig S5A, B, D, and E; ****P < 0.001 and **P < 0.01). Intriguingly, both cancer cell lines show increased levels of nesprin-2, when compared to RPE-1 (Fig S5C; ****P < 0.001). Nevertheless, these data clearly indicate that the LINC complex is altered in cancer cells, which correlates with their inability to correctly position centrosomes. Overall, our data strongly suggest that the LINC complex provides the cues for positioning centrosomes on the shortest nuclear axis during prophase.

### Dynein recruitment to the NE during early mitosis requires a functional LINC complex

Next, we sought to determine how LINC complex disruption affected centrosome positioning during prophase. While imaging DN-KASH–expressing cells, we frequently observed centrosome detachment from the NE during earlier stages of prophase (Fig 5H and I), which was never observed in cells expressing the ΔPPPL-KASH construct (Fig 5I) or unmanipulated RPE-1 cells. Curiously, this centrosome displacement from the NE decreased as cells entered mitosis (Fig 5I). These observations suggest that disruption of the LINC complex leads to a transient defect in centrosome–NE tethering during the G2-M transition, which is rescued as cells approach NEP. Previous studies have shown that a specific pool of NE-bound dynein is sufficient to tether centrosomes to the NE during prophase, by generating pulling forces on microtubules (Gönczy et al, 1999; Robinson et al, 1999), in a BicD2- (Splinter et al, 2010) or Nup133-dependent manner (Bolhy et al, 2011). Moreover, the LINC complex was previously shown to help maintain the centrosome–nucleus connection during neuronal migration in mice (Zhang et al, 2009). Thus, we decided to analyse whether the centrosome displacement we observed in DN-KASH cells could be due to a defective loading of dynein on the NE, triggered by disruption of the LINC complex. For this purpose, we assessed the localization of dynactin-1, a dynein adaptor

---

mock (I)- and Rap1* (J)-transfected RPE-1 cells, stably expressing H2B-GFP and RFP-tubulin, seeded on a 10-$\mu$m-wide micropatterned line. **(K, L)** Correlation between the angle centrosome–centrosome (x-axis) and the angle centrosome–nucleus (y-axis), at the moment of NEP for mock ((K); n = 26)- and Rap1* ((L); n = 31)-transfected cells. **(M, N)** Polar plot quantifying centrosome positioning relative to the longest nuclear axis at NEP for mock (M)- or Rap1* ((N); P = 0.583)-transfected RPE-1 cells. **(O)** Frames from a representative movie of a calyculin-A (CalA)-treated RPE-1 cell, stably expressing H2B-GFP and RFP-tubulin, seeded on a 10-$\mu$m-wide micropatterned line. **(P)** Cell membrane eccentricity of CalA-treated cells (orange; n = 22; ****P < 0.001). **(Q)** Correlation between the angle centrosome–centrosome (x-axis) and the angle centrosome–nucleus (y-axis) for cells treated with CalA. **(R)** Polar plot quantifying centrosome positioning relative to the longest nuclear axis at NEP for cells treated with CalA (P = 0.162). For all movies, time is in sec., and time zero corresponds to NEP. Scale bars, 10 $\mu$m.

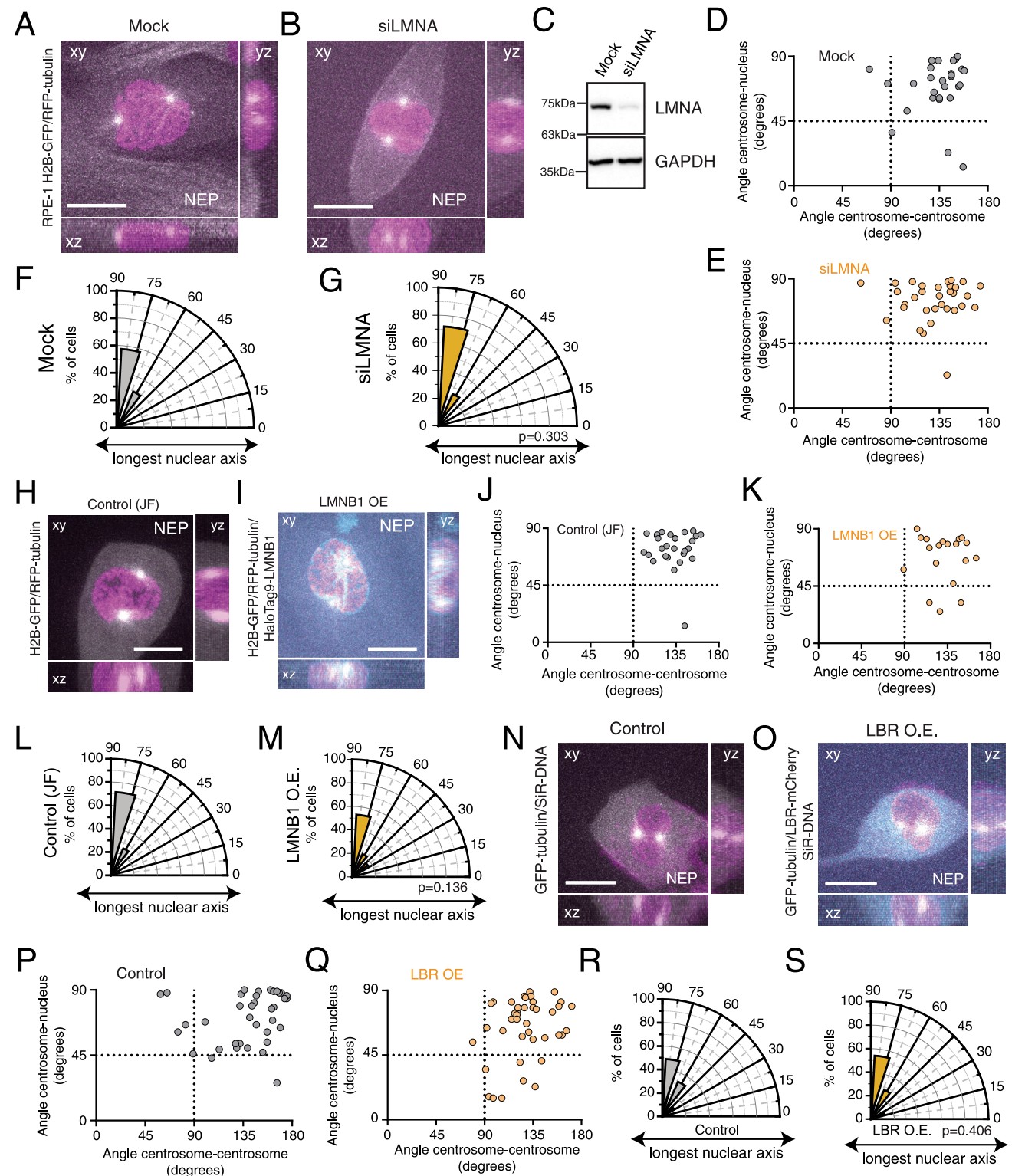

**Figure 3. Nuclear lamina components do not affect centrosome positioning.**
**(A, B)** Representative frames from movies of mock (A)- or lamin A–depleted ((B); siLMNA) RPE-1 cells at the moment of nuclear envelope permeabilization (NEP), stably expressing H2B-GFP and RFP-tubulin. **(C)** Western blot showing depletion efficiency of these cells. **(D, E)** Correlation between the angle centrosome–centrosome (x-axis) and the angle centrosome–nucleus (y-axis) at the moment of NEP, for mock ((D); n = 28)- and lamin A–depleted ((E); n = 32) cells. **(F, G)** Polar plots quantifying centrosome positioning relative to the longest nuclear axis at NEP for mock (F)- and LMNA-depleted RPE-1 cells ((G); P = 0.303). **(H, I)** Representative frame of the moment of NEP from movies of RPE-1 cells stably expressing H2B-GFP and RFP-tubulin ((H); control treated with JF647; n = 28) or overexpressing HaloTag9-LMNB1 ((I); LMNB1 OE; n = 19). **(J, K)** Correlation between the angle centrosome–centrosome (x-axis) and the angle centrosome–nucleus (y-axis; P = 0.216) at the moment of NEP, for control cells (J) and

that also localizes to the NE (Splinter et al, 2012), upon the expression of ΔPPPL-KASH or DN-KASH. Importantly, the expression of DN-KASH significantly decreased the levels of dynactin-1 on the NE, when compared to cells expressing ΔPPPL-KASH (Fig 5J and K; ****$P <$ 0.0001), while having no effect on the overall levels of cytoplasmic dynactin-1 (Fig 5L; n.s. $P = 0.079$) nor significantly changing the levels of another NE protein such as lamin B (Fig S4G and H; $P = 0.0823$). To further confirm the LINC complex requirement for dynactin loading on the NE, we proceeded to analyse its levels after SUN depletion (Fig 5M). We synchronized RPE-1 cells in late G2 using a CDK1 inhibitor (CDK1i; Ro-3306). Then, after inhibitor washout, we quantified the levels of dynactin on the NE at 3, 6, and 10 min, using immunofluorescence analysis (Fig 5N). As anticipated, cells depleted of SUN1 had decreased levels of dynactin at 3 min after release (**$P = 0.0086$). On the contrary, SUN2-depleted cells had significantly lower levels of dynactin at 10 min after release (****$P < 0.0001$). It should be noted that the effect of depleting individual SUN proteins on NE dynactin levels is not as severe as DN-KASH expression. This likely happens because the expression of the DN-KASH completely disrupts the interaction with both SUN proteins at the same time. On the contrary, with individual SUN depletion, the remaining SUN protein might still associate with nesprins to maintain a partial function. Interestingly, decreased levels of NE-associated dynactin-1 were also seen in both U2-OS and MDA-MB-468 cell lines when compared to RPE-1 cells (Fig S5F and G; ****$P < 0.0001$), which fits nicely with the decreased levels of LINC complex components (Fig S5A–E) and the defect in centrosome positioning (Fig 1) exhibited by these cancer cell lines. Taken together, our results indicate that an intact LINC complex is required for dynein localization at the NE during the G2-M transition, which then allows accurate centrosome positioning at NEP.

## Discussion

The manner in which centrosomes separate and position during prophase has direct implications on the efficiency of mitosis (Kaseda et al, 2012; Silkworth et al, 2012; Stiff et al, 2020). We previously reported that, during the G2-M transition, separating centrosomes exhibit a coordinated motion along the NE so that they are positioned on the shortest nuclear axis at the moment of NEP (Nunes et al, 2020). This centrosome–nucleus configuration occurs independently of the cortical cues that dictate spindle orientation in later stages of mitosis, because these cortical complexes are only assembled after NEP (Kiyomitsu & Cheeseman, 2012; Kotak et al, 2012). Importantly, this prophase configuration suggested that a nuclear signal could be providing the cues to regulate centrosome positioning during the early stages of mitosis. However, the molecular mechanism remained unclear. Here, we show that this

centrosome–nuclear axis orientation is a robust, LINC complex–dependent process in near-diploid, untransformed RPE-1 cells and that this mechanism is disrupted in U2-OS and MDA-MB-468 cell lines (Fig 6A and B).

In preparation for mitosis, most cells round up and adopt a spherical shape (Taubenberger et al, 2020). This event, driven by a combination of adhesion disassembly (Dao et al, 2009) and membrane retraction (Maddox & Burridge, 2003), is required to release the geometric constraints imposed by cell shape (Versaevel et al, 2012; Lancaster et al, 2013). In addition, this allows nuclear rotation during prophase (Nunes et al, 2020) and provides sufficient space for mitotic spindle assembly to occur without errors (Tse et al, 2012; Lancaster et al, 2013). Interestingly, we observed significant changes in the coordination between cell rounding and NEP in the cancer cell lines (Fig 1A–I), which resulted in an impairment of centrosome–nucleus axis repositioning. We hypothesized that abnormal cell rounding could be responsible for the defects in centrosome positioning that we observed in U2-OS and MDA-MB-468 cell lines. However, interfering with cell rounding in RPE-1 cells, either by delaying it or by accelerating it, did not disrupt accurate centrosome positioning. These observations rule out mitotic cell rounding as the main driver of centrosome positioning in early mitosis and reinforce the efficiency of centrosome–nucleus axis orientation in RPE-1 cells. In addition, they also suggest that the cues instructing centrosome positioning emanate from the nucleus. Using the nucleus as a positional cue during early mitosis has significant advantages. This would allow centrosomes to orient correctly under conditions where external cues are mostly absent, because of the disassembly of adhesion complexes in late G2 (Dao et al, 2009; Matthews et al, 2012), and before the cortical loading of the LGN-Gα1-NuMa complex, which occurs only in prometaphase (Kiyomitsu & Cheeseman, 2012; Kotak et al, 2012). In addition, this configuration would also enable the efficient assembly of a spindle scaffold (Nunes et al, 2020), ensuring a faster capture of kinetochores and mitotic progression (Kaseda et al, 2012; Silkworth et al, 2012; Nunes et al, 2020).

Many nuclear components could serve as potential candidates to influence centrosome behaviour. The LINC complex, because of its localization bridging the nucleus and the cytoplasm, is ideally placed to perform this function. Accordingly, it is involved in centrosome–nucleus tethering and nuclear migration in *Caenorhabditis elegans* (Fridolfsson & Starr, 2010) and myotubes (Cadot et al, 2012; Wilson & Holzbaur, 2012), and was also shown to affect multiple aspects of mitotic progression (Booth et al, 2019; Stiff et al, 2020; Belaadi et al, 2022). However, the molecular mechanism behind LINC complex–mediated centrosome positioning during the early stages of mitosis remained unclear. One likely possibility is that during late G2, the LINC complex could be directly involved in the recruitment of dynein to the NE. It is well established that

cells overexpressing LMNB1 (K). **(L)** Polar plots quantifying centrosome positioning relative to the longest nuclear axis at NEP for control (L) and LMNB1-overexpressing RPE-1 cells ((M); $P = 0.136$). **(N, O)** Representative frames of the moment of NEP from movies of control ((N); n = 37) or lamin B receptor (LBR)-mCherry–overexpressing ((O); n = 41) RPE-1 cells, stably expressing GFP-tubulin and treated with SiR-DNA, plated on fibronectin. **(P, Q)** Correlation between the angle centrosome–centrosome and the angle centrosome–nucleus at the moment of NEP, for control (P) and LBR-overexpressing (Q) cells. **(R, S)** Polar plots quantifying centrosome positioning relative to the longest nuclear axis at NEP for control (R) and LBR-overexpressing RPE-1 cells ((S); $P = 0.406$). For all images, scale bars = 10 $\mu$m. Source data are available for this figure.

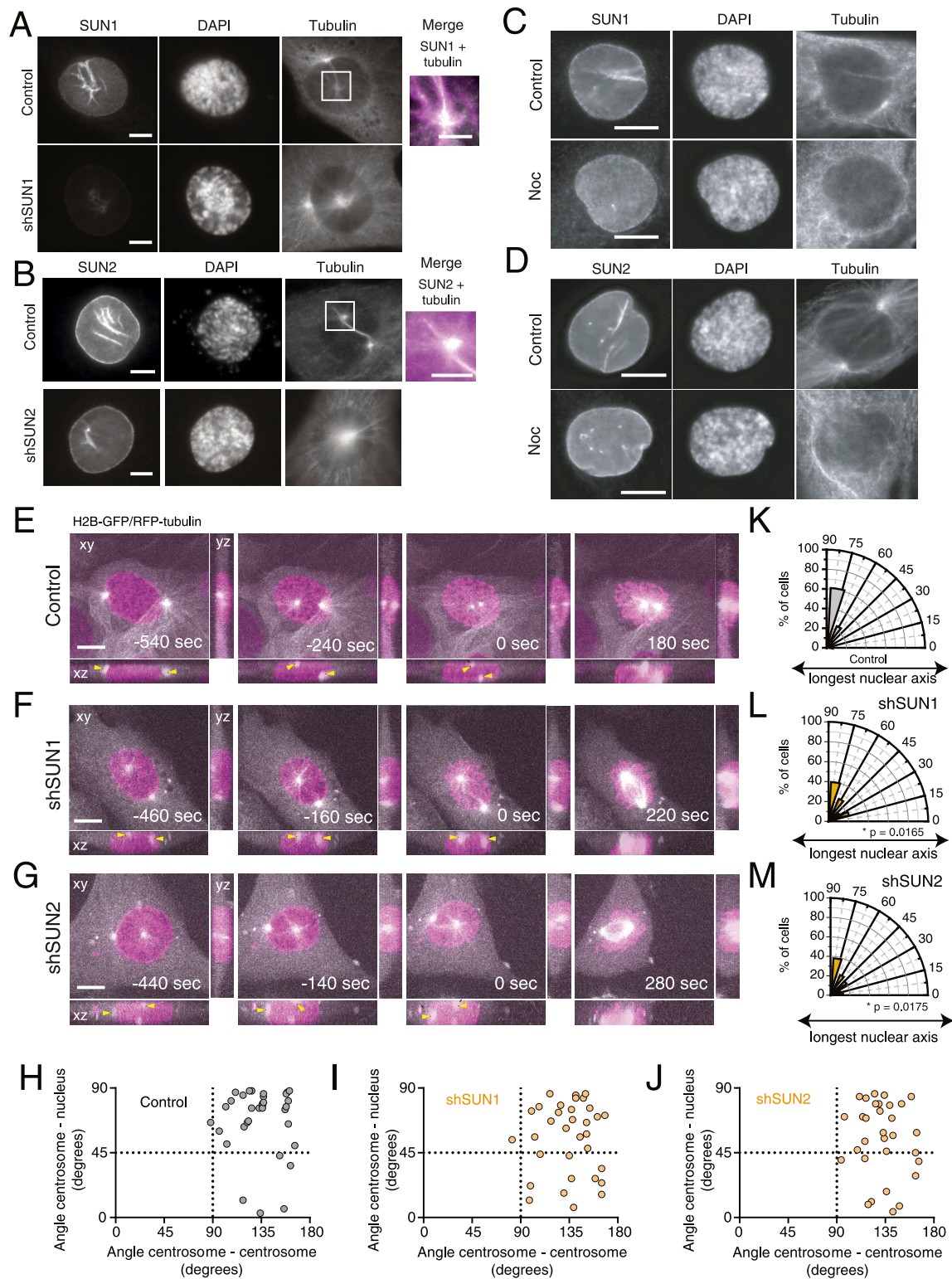

**Figure 4. SUN proteins are required for correct centrosome positioning.**
**(A)** Representative immunofluorescence images of control and shSUN1-treated cells, immunostained for SUN1. **(B)** Representative immunofluorescence images of control and shSUN2-treated cells, immunostained for SUN2. Note the co-localization between tubulin and the SUN proteins in the merged image inset.
**(C, D)** Representative immunofluorescence images of control and shSUN1-treated RPE-1 cells (C), or control and shSUN2-treated cells (D). Nocodazole (Noc) was added to the cells for 10 min, to induce microtubule depolymerization. Note the decrease in SUN1 and SUN2 staining between the centrosomes. **(E, F, G)** Representative frames from movies of control (E), shSUN1-treated (F), or shSUN2-treated (G) cells, stably expressing H2B-GFP and RFP-tubulin, seeded on fibronectin, during mitotic entry. Time is in sec. Time zero corresponds to nuclear envelope permeabilization (NEP). Yellow arrows indicate the centrosome position. **(H, I, J)** Correlation between the angle

during this stage, two parallel and independent nucleoporin-mediated pathways regulate dynein NE loading (Hu et al, 2013). Upon phosphorylation by CDK1 (Baffet et al, 2015), the nucleoporin RanBP2 binds to BicD2, leading to dynein recruitment (Splinter et al, 2010; Gallisà-Suñé et al, 2023). In addition, Nup133 recruits CENP-F, which then binds NudE/NudEL to load dynein on the NE (Bolhy et al, 2011). Once at the NE, dynein generates pulling forces on microtubules, which results in centrosome tethering to the nucleus, aiding centrosome separation (Raaijmakers et al, 2012) and facilitating NEP (Beaudouin et al, 2002; Salina et al, 2002). Here, we show that in untransformed RPE-1 cells, the LINC complex acts during late G2 to assist in NE-associated dynein loading (Fig 5) and provide spatial cues for centrosome positioning (Figs 4 and 5), in parallel with the NPC-mediated pathways. These observations fit nicely with a previous report showing a role of the LINC complex in centrosome movement during early mitosis (Stiff et al, 2020). In addition, LINC complex components SUN1 and SUN2, which are located on the inner nuclear membrane, were also shown to interfere with spindle assembly by delaying the removal of membranes from chromatin (Turgay et al, 2014; Belaadi et al, 2022). Together with our results, these observations indicate that the LINC complex acts on multiple levels during early spindle assembly. By assisting in the recruitment of NE-associated dynein, it ensures the accurate positioning of centrosomes on the shortest nuclear axis, to allow efficient assembly of the mitotic spindle. In line with this hypothesis, we and others have shown that disruption of the LINC complex, while unable to block mitotic entry, induces a severe delay in this process by decreasing the rate of cyclin B1 nuclear accumulation (Dantas et al, 2022) and interfering with spindle assembly efficiency (Turgay et al, 2014; Booth et al, 2019; Stiff et al, 2020). In parallel, by facilitating the removal of membranes from chromatin (Turgay et al, 2014), it also accelerates the capture of kinetochores by microtubules. Overall, this would result in a more efficient "search and capture" of chromosomes during early prometaphase, which, together with the "ring-like" distribution of chromosomes (Magidson et al, 2011), ensures a maximum exposure of kinetochores to microtubules and decreases the probability of generating erroneous kinetochore–microtubule attachments (Cimini et al, 2003; Silkworth & Cimini, 2012). Notably, mutations and the abnormal expression of LINC complex proteins have been implicated in a plethora of cancers (Sjöblom et al, 2006; Doherty et al, 2010; Matsumoto et al, 2015), suggesting a potential role in the maintenance of chromosomal stability. However, additional work is required to determine the exact molecular nature of the spatial cues that drive centrosome movement to the shortest nuclear axis. One hypothesis is that asymmetric localization of force-generating complexes on the NE could bias centrosome movement to a specific nuclear orientation. Accordingly, we observed an accumulation of SUN proteins in nuclear areas between the separating centrosomes, in a microtubule-dependent manner (Fig 4). One alternative hypothesis is that centrosomes generate pushing forces on the nucleus, sufficient to deform it and create a shortest axis. If so, then the LINC

complex would be necessary to ensure the timely loading of dynein on the NE so that centrosomes could tether to the nucleus and drive its deformation. Indeed, centrosome-mediated NE invaginations were previously reported to occur during the G2-M transition (Georgatos et al, 1997; Beaudouin et al, 2002; Salina et al, 2002; Turgay et al, 2014) and help in NEP. Overall, we propose a model (Fig 6) in which LINC complex–mediated loading of dynein at the NE dictates centrosome positioning at the shortest nuclear axis upon NEP and this is required to ensure an efficient spindle assembly in human cells.

# Materials and Methods

## Cell lines

RPE-1, U2-OS, and MDA-MB-468 cell lines were cultured in DMEM (Life Technologies) supplemented with 10% FBS (Life Technologies) and kept in culture in a 37°C humidified incubator with 5% $CO_2$. For MDA-MB-468 cells, media were also supplemented with GlutaMAX (Life Technologies). RPE parental, RPE H2B-GFP/mRFP-$\alpha$-tubulin, U2-OS parental, U2-OS H2B-GFP/mRFP-$\alpha$-tubulin, and MDA-MB-468 parental cell lines were already available in our laboratory. RPE-1 GFP-$\alpha$-tubulin, RPE-1 GFP-$\alpha$-tubulin/LBR-mCherry, and MDA-MB-468 H2B-GFP/mRFP-$\alpha$-tubulin cell lines were generated by transduction with lentiviral vectors containing the respective plasmids available in our laboratory (Table 1). For this purpose, we used HEK293T cells at a 50–70% confluence that were co-transfected with lentiviral packaging vectors (16.6 $\mu$g of Pax2, 5.6 $\mu$g of pMD2, and 22.3 $\mu$g of the plasmid of interest), using 30 $\mu$l of Lipofectamine 2000 (Life Technologies). ~48 h after transfection, the virus-containing supernatant was collected, filtered, and stored at –80°C. Cells were infected with the collected virus together with polybrene (1:1,000) in standard culture media for 24 h. Some days after the infection, the cells expressing the fluorescent tags were isolated by fluorescence-activated cell sorting (FACS; FACSAria II).

RPE-1 H2B-GFP/mRFP-$\alpha$-tubulin/shSUN1 and RPE-1 H2B-GFP/mRFP-$\alpha$-tubulin/shSUN2 were also generated by infecting cells, using commercially available lentiviral vectors encoding the desired shRNAs (shSUN1—TRCN0000133901—Target Sequence: CAGA-TACACTGCATCATCTTT; shSUN2—TRCN0000141958—Target Sequence: GCAAGACTCAGAAGACCTCT; MISSION shRNA, Sigma-Aldrich). Cells were then selected with puromycin (20 $\mu$g/ml; Merck Millipore) for 7 d. RPE-1 cells expressing the KASH constructs were generated via lentiviral infection. RPE-1 GFP-$\alpha$-tubulin cells were infected with viruses containing the mCherry-DN-KASH or mCherry-DN-KASH-$\Delta$PPPL plasmids, as described above. Doxycycline (5 $\mu$g/ml; Thermo Fisher Scientific) was added for 24 h to the culture media to stimulate the expression of the different KASH fusion proteins. After this period, the cells expressing the fluorescent tags were isolated by FACS and placed back in normal media for expansion. The

---

centrosome–centrosome (x-axis) and the angle centrosome–nucleus (y-axis) at the moment of NEP, for control ((H); n = 33), shSUN1-treated ((I); n = 33), and shSUN2-treated ((J); n = 31) cells. **(K, L, M)** Polar plots quantifying centrosome positioning relative to the longest nuclear axis at NEP for control (K), shSUN1-treated ((L); *$P$ = 0.0165), and shSUN2-treated ((M); *$P$ = 0.0175) cells. Yellow arrowheads indicate centrosome positions. All scale bars, 10 $\mu$m.

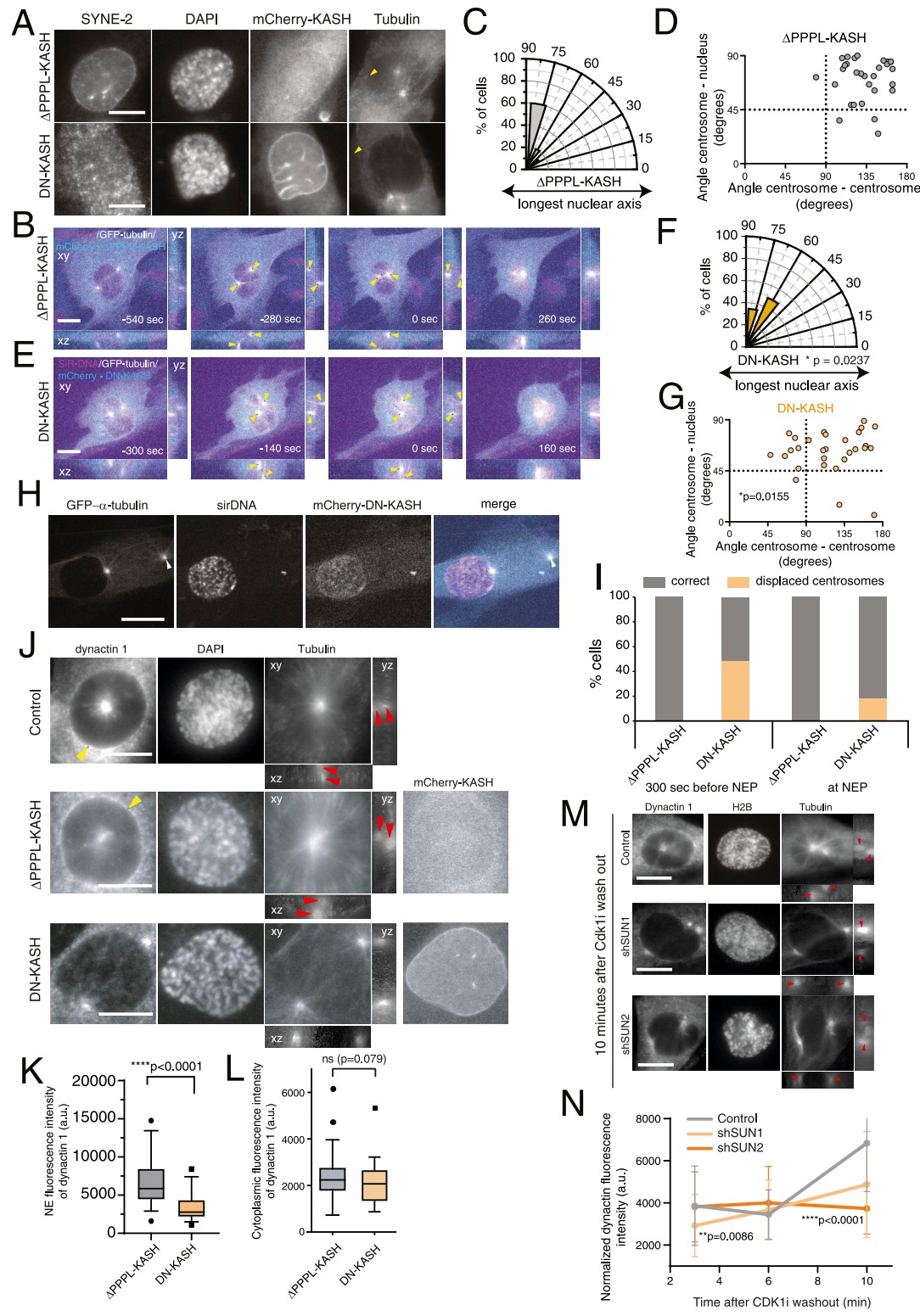

Figure 5. **Linker of nucleoskeleton and cytoskeleton complex is required for centrosome positioning on the shortest nuclear axis.**
**(A)** Representative immunofluorescence images of a ΔPPPL-KASH (top panel) and a DN-KASH (bottom panel) cell, immunostained for nesprin-2 (SYNE2). Note how the expression of DN-KASH displaces nesprin-2 from the NE (yellow arrows), as opposed to the expression of ΔPPPL-KASH. **(B, E)** Representative frames from control ((B); ΔPPPL-KASH) and DN-KASH (E)–treated cells, stably expressing tubulin-GFP and treated with SiR-DNA, seeded on fibronectin, during mitotic entry. Yellow arrowheads indicate centrosome position. **(C, F)** Polar plots quantifying centrosome positioning relative to the longest nuclear axis at nuclear envelope permeabilization (NEP) for RPE-1 expressing ΔPPPL-KASH ((C); n = 30) or DN-KASH ((F); n = 29; *P = 0.0237) cells. **(D, G)** Correlation between the angle centrosome–centrosome (x-axis) and the angle centrosome–nucleus (y-axis) at the moment of NEP for control ((D); ΔPPPL-KASH) or DN-KASH–expressing (G) cells. **(H)** Representative frame from a movie of a

addition of doxycycline was performed again 24 h before imaging. RPE-1 H2B-GFP/mRFP-α-tubulin/HaloTag9-lamin B1 cell line was generated by transiently transfecting a HaloTag9-lamin B1 plasmid (gift from Tom Misteli) using Lipofectamine 2000 (Life Technologies). Specifically, 5 µl of Lipofectamine 2000 and 0.6 µg of HaloTag9-lamin B1 plasmid were diluted separately and incubated in OptiMEM (Alfagene) for 30 min. The mixture was then added to confluent cells cultured and incubated for 6 h in reduced serum medium (DMEM with 5% FBS). Cells were selected using 400 µg/ml of G418 (Gibco, Life Technologies) for 14 d.

### Drug treatments

To visualize DNA, cells were incubated for at least 1 h with SiR-DNA (Spirochrome), at a final concentration of 10 nM, for a maximum of 4 h of imaging. To visualize HaloTag9 fluorescence, Janelia Fluor (JF) Ligand-647 (Promega) was added to the imaging media at a final concentration of 75 nM. To affect cell rounding, a Rho-associated protein kinase inhibitor (Y-27632) was used at 20 µM (Sigma-Aldrich) and cells were incubated with the drug 30–60 min before imaging. Calyculin-A (CalA, Abcam) was added to the cells during the imaging, at 20 µM for RPE-1 cells and 50 µM for U2-OS. To perturb microtubules, nocodazole (Noc, Sigma-Aldrich) was used at 3.3 µM. Control cells were treated with the corresponding volume of DMSO (Sigma-Aldrich).

### Transfections

Cells were transfected with the plasmid pRK5-Rap1[Q63E] (Rap1*; a gift from Jean de Gunzburg) for 48 h as described before for plasmid transfections. Control cells were transfected with Lipofectamine 2000 (Invitrogen) only, in the same conditions. To deplete lamin A, RPE-1 cells were transfected with siRNAs using Lipofectamine RNAiMax (Life Technologies). Specifically, 5 µl of Lipofectamine RNAiMax and 20 nM of each siRNA were diluted separately and then incubated in OptiMEM (Alfagene) for 30 min. The mixture was then added to confluent cells cultured and incubated for 6 h in reduced serum medium (DMEM with 5% FBS). Commercial ON-TARGETplus SMARTpool siRNAs (Dharmacon) were used. Commercial ON-TARGETplus SMARTpool Non-targeting siRNAs and mock transfections were used as controls. Cells were analysed 72 h after transfection, and protein depletion efficiency was verified by immunoblotting.

### Western blotting

Cell extracts were collected after trypsinization and centrifuged at 150g for 5 min, washed in PBS twice, and resuspended in 30–50 µl of lysis buffer (50 mM Tris–HCl, pH 7.4, 150 mM NaCl, 1 mM EGTA, 0.5%

NP-40, and 0.1% Triton X-100), 1:50 protease inhibitor (cOmplete Tablets EASYpack; 04693116001; Roche), and 1:100 PMSF. Cells were kept on ice for 30 min, then flash-frozen in liquid nitrogen twice. DNA was pelleted after centrifugation at 21,000g for 8 min at 4°C, the supernatant was collected, and the protein concentration was determined using the Bradford protein assay (Bio-Rad). Proteins were run on a 10% SDS–PAGE (30 µg/lane) and transferred using a semi-dry blotting system (Trans-Blot Turbo System; Bio-Rad) for 10 min at 25 V, with constant amperage. Next, the membranes were blocked with 5% milk in PBS with 0.05% Tween-20 (PBS-T) for 1 h, at RT. The primary antibodies used were anti-lamin A (C-terminal; 1:1,000; L1293; Sigma-Aldrich), anti-SUN1 (1:500; MABT892; Merck Millipore), anti-SUN2 (1:500; MABT880; Merck Millipore), anti-GAPDH (1:20,000; 60004-1-Ig; Proteintech), and anti-β-tubulin (1:5,000; Ab6046; Abcam). All primary antibodies were incubated overnight at 4°C with shaking. After three washes in PBS-T, the membranes were incubated with the secondary antibody for 1 h, at RT. The secondary antibodies used were anti-mouse HRP and anti-rabbit HRP, at 1:5,000. After three washes with PBS-T, the detection was performed with Clarity Western ECL Substrate (Bio-Rad). Acquisition of blots was performed with a Bio-Rad ChemiDoc XRS system using IMAGE LAB software.

### Immunofluorescence

Cells were seeded the day before the experiment on coverslips coated with FBN (25 µg/ml; F1141; Sigma-Aldrich). When necessary, cells were treated with the appropriate compounds, fixed with 4% PFA in cytoskeleton buffer (1.25 M NaCl, 1 M KCl, 125 mM Na$_2$HPO$_4$, 250 mM KH$_2$PO$_4$, 250 mM EGTA, 250 mM MgCl$_2$, 250 mM PIPES, and 500 mM glucose, pH 6.1) for 10 min at RT, and then extracted in PBS with 0.5% Triton X-100 (Sigma-Aldrich) for 5 min (or 30 min when using the antibody against dynactin-1). Coverslips were then blocked using 10% FBS in 0.1% Triton X-100 for 30 min, at RT. These coverslips were afterwards incubated with the following primary antibodies: rabbit anti-SUN1 (1:200; HPA008346; Sigma-Aldrich), rabbit anti-SUN2 (1:200; HPA001209; Sigma-Aldrich), mouse anti-nesprin-2 (1:200; sc-398616; Santa Cruz Biotechnology), mouse α-tubulin (α-Tubulin B-5-1-2; 1:1,000; 32–2,500; Sigma-Aldrich), rabbit β-tubulin (1:1,000; Ab6046; Abcam), mouse anti-lamin A+C (1:500; Ab8984; Abcam), rabbit anti-lamin B1 (1:1,000; ab16048; Abcam), rabbit α-dynactin-1 (1:200; PA5-21289; Invitrogen), rabbit anti-nesprin-1 (1:200; PA5-82666; Invitrogen), and anti-LBR (1:500; HPA062236; Atlas Antibodies) in blocking solution. Primary antibody incubation was usually performed for 1 h at RT, except when probing cells with the anti-nesprin-1, anti-nesprin-2, and anti-dynactin-1 antibodies, in which case incubation was completed

DN-KASH–expressing cell, stably expressing tubulin-GFP and treated with SiR-DNA, highlighting centrosome detachment from the NE (white arrowheads). **(I)** Quantification of centrosome displacement from the NE during mitotic entry for ΔPPPL-KASH– or DN-KASH–expressing cells before (−300 s) and at NEP. **(J)** Representative immunofluorescence images of control (top panel), ΔPPPL-KASH (middle panel), or DN-KASH (bottom panel) cells, labelled for the dynein adaptor, dynactin-1. Please note the decreased dynactin-1 signal at the NE of DN-KASH–expressing cells, when compared to control or ΔPPPL-KASH cells (yellow arrowheads). Red arrowheads indicate centrosomes positioned above and beneath the nucleus (lateral projections). **(K, L)** Quantification of the normalized fluorescence intensity signal of dynactin-1 signal on the NE ((K); ***P < 0.001) and cytoplasm ((L); P = 0.079) of ΔPPPL-KASH (n = 36) or DN-KASH (n = 35) cells. **(M)** Representative immunofluorescence images of control (top panel), SUN1 depleted (middle panel) and SUN2 depleted (bottom panel) RPE-1 cells, showing dynactin-1 accumulation on the NE. Red arrowheads indicate centrosomes. **(N)** Quantification of dynactin-1 levels on the NE following CDK1i washout for the different experimental groups. **P = 0.086, ****P < 0.0001. Time is in sec, and time zero corresponds to NEP. Scale bars, 10 µm.

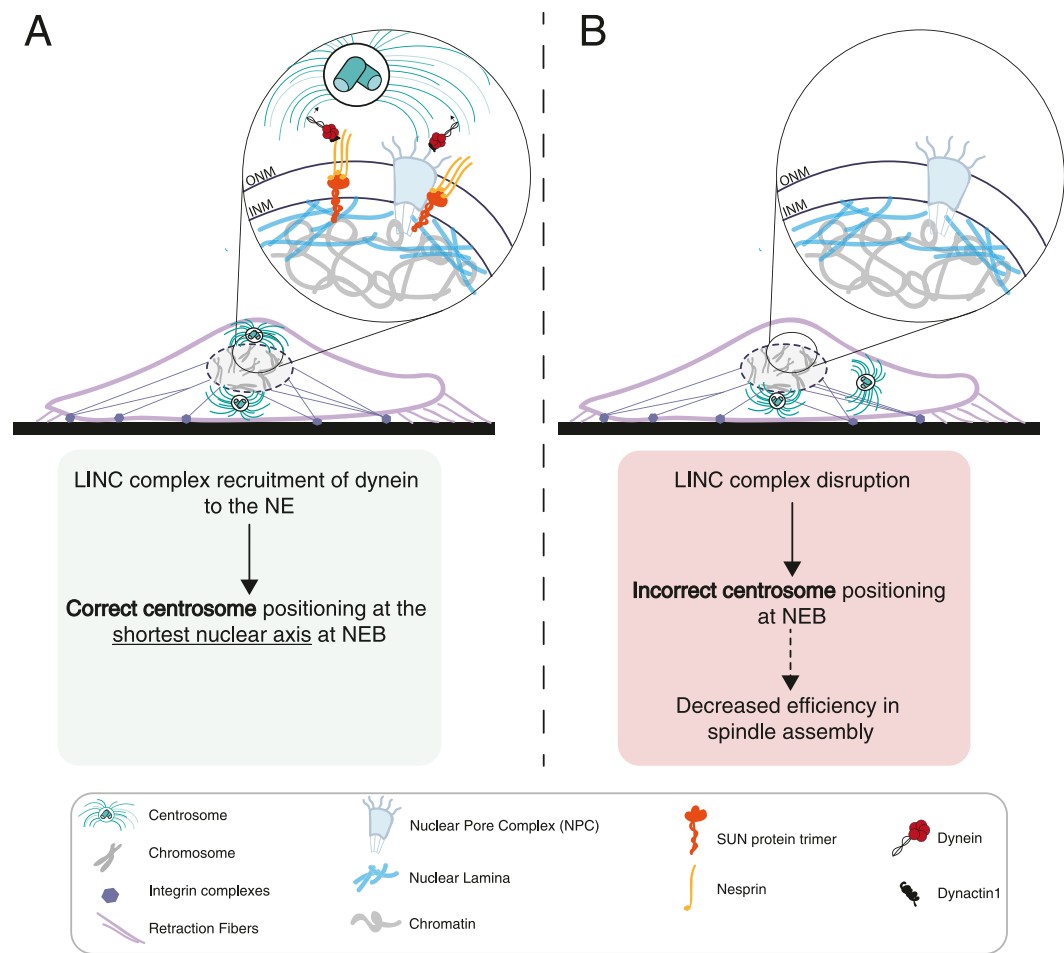

**Figure 6. Proposed model for the linker of nucleoskeleton and cytoskeleton (LINC) complex–dependent centrosome positioning.**
**(A)** In normal conditions, the LINC complex allows timely loading of dynein on the NE, leading to correct centrosome positioning. This allows efficient mitotic spindle assembly and chromosome segregation. **(B)** Upon LINC complex disruption, the loading of dynein on the NE is delayed, leading to incorrect centrosome positioning. Consequently, mitotic spindle assembly is perturbed, which could result in chromosomal instability, under the appropriate genetic background.

**Table 1.   List of plasmids used in this work.**

| Plasmid | Type | Origin |
|---|---|---|
| pLKO.1 H2B-GFP | Lentiviral | Addgene |
| pRRL-mRFP-α-tubulin | Lentiviral | In-house |
| Rap1* | Transient expression | Gift from Jean de Gunzburg |
| α-Tubulin-GFP | Lentiviral | In-house |
| LBR-mCherry | Lentiviral | Gift from Stephen Royle |
| HaloTag9-lamin B1 | Transient expression | Gift from Tom Misteli |
| pLKO.1 shSUN1-puro | Lentiviral | Sigma-Aldrich |
| pLKO.1 shSUN2-puro | Lentiviral | Sigma-Aldrich |
| pinducer 20 DN-KASH | Lentiviral | #125554; Addgene |
| pinducer 20 DN-KASH-ΔPPPL | Lentiviral | # 129280; Addgene |

overnight at 4°C. Coverslips were washed with PBS/0.1% Triton X three times (5 min each) and incubated with the secondary antibodies (1:2,000; Alexa Fluor–conjugated; Invitrogen), at RT for 1 h.

When appropriate, DAPI (1 μg/ml; Invitrogen) was added to the secondary antibody mixture to stain DNA. Finally, coverslips were washed three times in PBS with 0.1% Triton X-100 and once in PBS,

and sealed on a glass slide using mounting medium (20 nM Tris, pH 8, 0.5 N-propyl gallate, and 90% glycerol). Images were acquired using an AxioImager Z1 (63x, plan oil differential interface contrast objective lens, 1.4NA; all from Carl Zeiss), which is coupled to a CCD camera (ORCA-R2; Hamamatsu Photonics) using Zen software (Carl Zeiss).

### Micropatterning

Micropatterning was performed using a deep-UV light technique to normalize cell shape and adhesion area, as previously described (Azioune et al, 2009). Glass coverslips (square 22 × 22 mm, 1.5, VWR; or round 25 mm, 1.5; Thermo Fisher Scientific) were activated with plasma (Zepto Plasma System, Diener Electronic) for 2 min. After plasma treatment, coverslips were incubated with 0.2 mg/ml PLL(20)-g [3,5]-PEG(2) (SuSoS) in 10 mM Hepes at pH 7.4, for 1 h, at RT. Coverslips were washed three times with water, and left to dry before being placed on a synthetic quartz photomask (Delta Mask), previously activated with deep-UV light (PSD-UV; Novascan Technologies) for 5 min, using 3 µl of Milli-Q water to seal it to the mask. The coverslips were then irradiated through the photomask with the UV lamp for 5 min and left to dry before being incubated with FBN (25 µg/ml; F1141; Sigma-Aldrich), in 100 mM NaHCO3 at pH 8.6, for 30 min, at RT. Whenever possible, 5 µg/ml Alexa Fluor 647–conjugated fibrinogen (Thermo Fisher Scientific) was added to the FBN mix in order to visualize the pattern surfaces. Cells were added to the freshly incubated coverslips and allowed to spread for 15 min, before removing excess cells and new culture medium was added, and cells were left to fully adhere for another 12–16 h.

### Time-lapse microscopy

Cells seeded on patterned or non-patterned surfaces are placed in Leibovitz's L15 medium (Life Technologies), supplemented with 10% FBS and Antibiotic/Antimycotic Solution 100X (AAS; Life Technologies) right before imaging, alongside the corresponding drugs, where indicated. Live-cell imaging experiments were performed using temperature-controlled Nikon TE2000 microscopes equipped with a modified Yokogawa CSU-X1 spinning-disc head (Yokogawa Electric), an electron multiplying iXon+ DU-897 EMCCD camera (Andor), and a filter wheel. Three laser lines were used to excite 488, 561, and 647 nm, and all the experiments were done with immersion oil using a 60x 1.4NA Plan-Apo DIC (Nikon). Image acquisition was controlled by NIS-Elements AR software. Images were obtained with 17 z-stacks (0.5 µm step) with a 20-s interval when assessing centrosome positioning during mitotic entry.

### MATLAB custom algorithm for centrosome tracking

Analysis of centrosome positioning and behaviour during mitotic entry was performed using a custom-designed MATLAB (v2018b; The MathWorks, Inc.) script (Castro et al, 2020) that contains a specialized workflow previously optimized for centrosome tracking, together with the reconstruction of both cellular and nuclear membranes in a 3D space. A pixel size of 0.176 µm and a z-step of 0.5 µm were taken into consideration. Error correction methods were employed in cases where the standard automatic method was unable to correctly detect the two centrosomes and membranes. These correction methods involved, amongst others, manual centrosome position adjustment in all three coordinates (x, y, and z) for each frame, and threshold correction for membrane reconstruction. From these membrane reconstructions, the algorithm was able to extract cell and nuclear major axis, as well as cell and nuclear membrane eccentricity and irregularity levels. From the coordinates obtained for centrosomes and with the nuclear and cell axis computed, the tool was able to calculate the angle between the two centrosomes that passed through the centroid of the nucleus (angle centrosome–centrosome), the angle of the centrosome axis relative to the long nuclear axis (angle centrosome–nucleus), and the angle formed between the longest cell axis and the longest nuclear axis, for each frame/time point.

### Quantification of nuclear solidity

Nuclear solidity was quantified using the shape descriptor plugin from ImageJ. Briefly, nuclei are outlined using the polygon tool and the nuclear area is measured. To calculate nuclear solidity, the nuclear area is then divided by the corresponding convex hull area. Irregular nuclei will typically show a lower nuclear solidity value.

### Quantitative image analysis of dynactin-1 and LINC complex protein levels

For the quantification of dynactin-1, nesprin, and SUN protein levels at the NE, images were analysed using ImageJ. A sum projection of three z-slices encompassing the central region of the nucleus was employed in all measurements. On the sum-projected image, a segmented line (smoothened by a spline fit) of a defined width (w1) was drawn along the NE, and the transverse-averaged fluorescence signal (S1), which contains signal and background, was measured. The S1 (transverse average) is directly retrieved using Ctrl-M or Ctrl-K in ImageJ. A second equivalent measurement (S2) was done using the same line after increasing its width to w2. Although the signal of interest remains the same in the dilated line, the background increases by the factor w2/w1, the knowledge of which allows retrieval of I(r), the background-corrected profile, using the following:

$$I(r) = \frac{w_1 w_2}{w_2 - w_1} (S_2(r) - S_1(r))$$

Line width w1 should be large enough to fully encompass the signal of interest, whereas w2 should be at least 20% larger than w1 but small enough to avoid the inclusion of extraneous signal from non-NE sources. In the particular quantification done in this study, the intensity profile (i.e., the r-dependence) was irrelevant, so I(r) was integrated along the full length of the curve and divided by the line length (or, equivalently, the line "area").

### Statistical analysis

At least three independent experiments were used for statistical analysis. Sample sizes and number or replicates are indicated in each figure legend. The normality of the samples was assessed using the Kolmogorov–Smirnov test. Statistical analysis of multiple

group comparison was performed using a parametric one-way ANOVA when the samples had a normal distribution. Otherwise, multiple group comparison was done using a non-parametric ANOVA (Kruskal–Wallis). Multiple comparisons were analysed using either the post-hoc Student–Newman–Keuls (parametric) or Dunn's (non-parametric) tests. When comparing two experimental groups only, a parametric $t$ test was used when the sample had a normal distribution, or a non-parametric Mann–Whitney test was used for samples without normal distribution. Comparison of multiple time-course datasets was done using repeated-measures ANOVA, when the samples had a normal distribution. Otherwise, group comparison was done using repeated-measures ANOVA on ranks. No power calculations were used. All statistical analyses were performed using the GraphPad Prism (Dotmatics). When comparing proportions between two populations, a z-score was calculated.

## Supplementary Information

## Acknowledgements

This work was funded by Portuguese funds through FCT—Fundação para a Ciência e a Tecnologia/Ministério da Ciência, Tecnologia e Ensino Superior in the framework of the project PTDC/BIA-CEL/6740/2020. JT Lima is supported by grant SFRH/BD/147169/2019 from FCT. The authors thank Helder Maiato for critical reading of the article and access to essential equipment for live-cell imaging. The authors would like to thank Tom Misteli for the HaloTag9-lamin B1 plasmid, Jean de Gunzburg for the pRK5-Rap1[Q63E] construct, and Stephen Royle for the plasmid pWPT LBR-mCherry.

### Author Contributions

JT Lima: conceptualization, data curation, formal analysis, investigation, visualization, methodology, and writing—original draft, review, and editing.
AJ Pereira: resources, software, formal analysis, visualization, methodology, and writing—review and editing.
JG Ferreira: conceptualization, resources, formal analysis, supervision, funding acquisition, validation, visualization, methodology, project administration, and writing—original draft, review, and editing.

### Conflict of Interest Statement

The authors declare that they have no conflict of interest.

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
