## [Reviewer comments · Life Science Alliance]

Life Science Alliance

The LINC complex ensures accurate centrosome positioning during prophase

Joana Lima, Antonio Pereira, and Jorge Ferreira

DOI: <https://doi.org/10.26508/lsa.202302404>

Corresponding author(s): Jorge Ferreira, *i3S - Instituto de Investigação e Inovação em Saúde, Universidade do Porto*

Review Timeline:

Submission Date:	2023-09-28
Editorial Decision:	2023-09-29
Revision Received:	2023-11-22
Editorial Decision:	2023-12-18
Revision Received:	2024-01-03
Accepted:	2024-01-04

Transaction Report:

Please note that the manuscript was reviewed at *Review Commons* and these reports were taken into account in the decision-making process at *Life Science Alliance*.

Reviews

Review #1

****Summary:**** in this study, Lima and colleagues, investigate the mechanisms controlling the position of the two centrosomes at nuclear envelope breakdown. The authors show that in the non-cancerous human epithelial RPE1-cell line, the centrosomes are generally positioned in the short axis of the nucleus; in contrast in two cancer cell lines, they did not find an equivalent pattern. When the authors set out to identify potential molecular players required for this positioning, they find that the LINC complex is required, possibly by recruiting dynein to the nuclear membrane. Finally, the authors show that disruption of the LINC complex is associated with chromosome segregation errors.

****Major Comments:****

In general, the presented experiments are of excellent technical quality. The main conclusions of the manuscript, are however, not always well supported by the experimental data. They should be either interpreted more cautiously or supported by additional experimental evidence. I highlight these here, using the main conclusions of the abstract.

1. "We show that in untransformed cells, centrosome positioning is regulated by a nuclear signal, independently of external cues. »

The authors conclude based on three cell lines that the centrosome positioning mechanisms is present in non-transformed cells and not in cancerous cells. The authors have, however, only analysed 1 non-cancerous cell line, and they compare cells originating from vastly different tissues (retina, bones and breast) and origins (epithelial vs. mesenchymal cartilage cells). Such a general statement is not possible, without a systematic comparison of several healthy cells vs cancerous cells from the same tissue.

2. "This nuclear mechanism relies on the linker of nucleoskeleton and cytoskeleton (LINC) complex that controls the loading of dynein on the nuclear envelope (NE), providing spatial cues for robust centrosome positioning on the shortest nuclear axis, prior to nuclear envelope permeabilization (NEP). »

While the data showing that centrosome positioning depends on the LINC complex is solid and robust, some of the "negative" examples identified by the authors are less convincing. One the process the authors study is cell rounding. Based on the fact that Rap1 transfection or treatment with Calyculin A does not lead to differences that are statistically different, the authors conclude that cell rounding is not involved. However, absence of statistical difference does not mean that there is no difference. Indeed, when comparing the raw data in Figure 2L and 2Q to the positive hit shSun2 in Figure 4J, one could conclude that cell rounding does make a difference, and that this statistical difference would emerge if the authors would count a high number of cells. Therefore the authors should interpret these results in a more differentiated manner, and also instead of just stating non-significant, state also the real p-values for the different experiment.

The second major concerns emerges when looking at the data in Figure 5, when the authors test for the abundance of the dynein complex on the nuclear envelope in cells treated with DPPPL-KASH or DN-KASH. Yes, there is a statistical difference, but the absolute difference is tiny (I estimated a normalized intensity of 1.44 vs 1.35). This is a difference of less than 10%. How do the authors think that such a small change in dynein could have such a strong effect on centrosome positioning? Would a partial dynactin depletion by 10% give an equivalent result? Does the depletion of other proteins involved in the late recruitment of dynein at the NE also affect centrosome positioning?

3. « Moreover, we demonstrate this mechanism is altered in cancer cells, leading to increased chromosome segregation errors. »

Here the authors infer that the identified mechanism is absent in cancer cells and that its absence contributes to chromosome segregation errors. Both conclusions are not supported by the presented data. First, the authors did not test whether any members of the LINC complex or dynactin is present at lower levels on the nuclear membranes of the cancer cells. Such a direct validation would be essential to make such a strong statement. Second, the authors conclude that this mechanism prevents chromosome segregation errors, based on the fact that depletion or impairment of the LINC complex (shSUN1, shSUN2, DN-KASH) results in chromosome segregation errors. These perturbations lead, however, as noted by the authors themselves to pleiotropic effects, including insufficient retraction of nuclear membrane, which will can all contribute to chromosome segregation

errors. It is therefore impossible to estimate the contribution of the centrosome positioning mechanism to these segregation errors using this type of perturbations. One could even argue that this mechanism might not be that important, since depletion of SUN2, which also impairs centrosome positioning has no significant effect on chromosome segregation.

****Minor comments:****

The author state in the Material and methods that all the figure legends contain the number of replicates. This is, however, not the case, the authors only indicate the total number of analyzed cells.

****Referees cross-commenting****

I agree that all three reviewers come to similar conclusions - strong technical quality, novel results and concepts, but some limitations due to lack of precise tools or the limited number of model cell lines investigated.

I recommend that the authors prioritize which are the suggested experiments that could be done within a few months, and otherwise rephrase their conclusions in less general terms.

2. Significance:

Significance (Required)

This study establishes for the first time that some cell lines set up the mitotic spindle at predefined positions of the nucleus and they identify a first molecular complex controlling this complex.

The strength of this study is the high technical quality of the data - a limitation is the over-interpretation of the current data (see major comments), and the fact that the authors do not have a tool that specifically only disrupts centrosome positioning, which would allow them to probe the importance of this mechanism.

3. How much time do you estimate the authors will need to complete the suggested revisions:

Review #2

A nuclear signal in prophase determines centrosome positioning and ensures efficient mitotic spindle assembly. Lima and Ferreira investigate in this manuscript the regulation of centrosome positioning in early mitosis. The authors first analyze the position of the two centrosomes either relative to the cell length axis or the shortest or longest axis of the nucleus and describe differences between RPE1, U2OS, and MDA-MB cells. Next, they analyze whether mitotic cell rounding determines the position of centrosomes; however, delayed cortical retraction (Rho-kinase inhibition), adhesion disassembly inhibition (Rap1Q63E), and inducing premature rounding (CalA) did not impact centrosome positioning in RPE1, U2OS, and MDM-MB cells. In addition, the nuclear lamina and LBR were also not required for centrosome positioning on the shortest nuclear axis. In contrast, depletion of SUN1 or SUN2 and overexpression of a dominant-negative DN-KASH affected the nuclear positioning of centrosomes in RPE1 cells. Finally, the authors analyze whether the LINC complex impacts mitotic fidelity. This is indeed the case when SUN1 is depleted, but it is not the case for SUN2 depletion or DN-KASH overexpression. This difference between LINC complex components is not discussed in the manuscript. Since SUN1, SUN2, and DN-KASH affect centrosome positioning in a similar way (Figs. 4 and 5), the chromosome segregation defect in SUN1-depleted cells is most likely not caused by a centrosome position defect but probably by another defect caused by SUN1 depletion.

****Major comments****

1. Figure 1 is insufficiently explained. The authors have to describe in an understandable way how they measured centrosome-centrosome angle and centrosome-nucleus angle. They should show a cartoon in which these angles are clearly shown. The small cartoons in Fig. 1C are not helpful at all; they are also not explained. The authors should explain the meaning of the black dots (are these centrosomes?) and the even smaller dots. The short nuclear axis should be indicated, e.g., by a red line.

2. On the first page of the manuscript: "Consequently, at the NEP, centrosomes are positioned on the shortest nuclear axis (Fig. 1C) as can be seen in Fig. 1A. This means that the centrosome-nucleus angle relative to the

shortest nuclear axis should be 0. However, in Fig. 1C, this angle is between 45 and 90 degrees. This is also the case for Fig. 1D. Please clarify.

3. I find it confusing that in Fig. 1, depending on the subfigure, the short or longest nuclear axis is used as a reference point: Fig. 1C: shortest; D: shortest; F: shortest; G: longest; I: shortest; J: longest. Thus, even within the same cell line, the reference point is changing. What is the rationale for this variation?

4. Fig. 4K, L, M: in figure, y-axis: "shortest nuclear axis". In legend: "relative to the longest nuclear axis". I guess the longest nuclear axis is correct. Same in Fig. 5D and E. Fig. 5C lacks the WT control.

5. The cells in Fig. 5J are not comparable: one has a monopolar spindle, the other a bipolar. The authors need some other NE protein as a control to show that the reduction of dynein by DN-KASH is a specific defect and not a broad impact on the NE. The dynein data in Figs. 5J-L need to be extended to SUN1/2.

6. The title of the paper is misleading: the authors do not provide any indication for a nuclear signal in prophase that determines centrosome positioning.

****Minor comment****

1. It would make sense to use the same time scale in Figs. 1A and B (either min.sec. or sec.) to allow direct comparison.

2. 2nd section: Mitotic cell rounding "The authors state: Given that cancer cells failed... I would be careful with this generalization; only one cancer cell was used in this study.

3. The authors say: "However, they did not place the centrosomes at the shortest nuclear axis (Figure 4K-M)." Centrosomes are still on the shortest nuclear axis but not as frequent as in control.

4. The white color in Fig. 6B cannot be seen and needs to be changed to something else.

5. The paper has neither line nor page numbers.

****Referees cross-commenting****

My comments are more or less reflected by the comments and concerns of reviewer 1 (only one cancer cell line; the role of the LINC complex). This reduced the impact of this manuscript that is certainly interesting and has novel aspects.

2. Significance:

The manuscript analyzes an early step in spindle assembly: the positioning of the two centrosomes on the NE. As such, the paper is interesting and important. They exclude cell rounding and lamin disassembly as mechanisms for centrosome positioning. The SUN1/2 and KASH data on centrosome positioning are convincing, and they provide a novel finding on the function of the LINC complex in centrosome positioning, probably via dynein recruitment to the NE. It remains unclear whether LINC recruits dynein directly or functions via one of the two known dynein/NE recruitment pathways. LINC-dynein at the NE binds centrosome microtubules and dynein pulls them towards the NE. However, how LINC-dynein spatially positions centrosomes relative to the short axis of the nucleus remains unclear (dynein uniformly decorates the NE (Fig. 5J)). The data on chromosome missegregation are not so clear because the defect only occurs in SUN1-depleted cells. Thus, this phenotype indicates most likely a function of SUN1 but not the LINC complex and is probably not related to centrosome positioning since all LINC components affect centrosome positioning. The paper falls short in explaining how parameters were measured and contains mistakes in the figures, as outlined above. The paper lacks a coherent story (a little bit on cancer, some negative data, LINC-dynein, but it stops on the surface).

It will be relatively easy to improve some aspects of the manuscript (explaining the angles, correcting the figures: one week). Measuring dynein at the NE in SUN1/2-depleted cells is also easy to do (1-2 months). To get more mechanistic insights into how LINC-dynein positions centrosomes probably will not be possible during revision time.

Review #3

****Summary:**** Centrosomes separate early in mitosis to allow faithful spindle assembly and chromatin segregation. In the current study the authors show that in RPE-1 cells the separated centrosomes typically position each other along the shorter axis of the nucleus while in cancer derived U2OS and MDA-MB-486 this is

rather random. Mitotic cell rounding is not causal for this effect. Rather, the LINC (linker of nucleoskeleton and cytoskeleton) complex, a protein complex spanning both membranes of the nuclear envelope, is required for this. The data indicate that this is dynactin1 recruitment to the nuclear envelope. The work suggest that proper arrangement of the centrosomes along the short nuclear axis via the LINC complex contributes to chromatin segregation fidelity in RPE-1 cells.

****Major comments:**** The data, derived mostly by live cell imaging of cell culture lines, are of very high quality, carefully controlled and analyzed. They fully support the claims of the study and are well presented, both in text and figures. Statistical analysis seems adequate, but since the authors show different kinds of data sets including time series and use several kinds of statistical tests, it would make sense to indicate the test used for each p-value in all the figure legends. I have no major criticism or experiments to suggest.

****Minor comments:****

1. Figure legends are quite repetitive and could be shortened. E.g. in Fig. 1 the description for E, F, H and I repeats what has been explained for B and C. Same applies between figure legends. The authors might refer to previous legends if the analysis was done in a similar way.
2. How is nuclear solidity defined and analyzed in Fig S3D?
3. The references to Fig S3 in figure legend 3 ("see Fig S3") do not enlighten the message and could be removed. The same applies to Fig5 - here it is not clear why the author refer to Fig S4.
4. Fig. 3: I suggest to quantify the lamin B1 and LBR overexpression levels.
5. Fig. 5: Consider reordering the panel: Start with the current panel C (as in the text) as it is the necessary control prior to the experimental data.
6. Fig 5 I: what means "before"? Can the authors give a time window they use for analysis.
7. Page 20: "... shortest nuclear axis (Fig. 1C, 5D-G; n.s. - not significant). However, DN-KASH-expressing cells showed compromised separation and positioning of centrosome (Fig. 5D-G, * p=0.0155 and * p=0.0237, respectively). - rather point to the specific panels, i.e. Fig. 1C, 5D and F as well as and Fig. 5E and G.
8. Fig 6B. The DN- KASH bars are on my pdf not visible - use a darker grey
9. Fig S6, albeit mentioned in the text, is not included in the supplementary info.
10. Material and Methods: in general very clear and carefully written
 - a. GlutaMAX instead of GlutaMAXE (page 29)
 - b. What means "as described previously"? No reference is given. Do you refer to the upper part of the method section? (page 30)
 - c. 20 nM HEPES should most probably read 20 mM (page 32)
 - d. "1:50 protease inhibitor; 1:100 Phenylmethylsulfonyl fluoride" - which protease inhibitor (mixture)? Rather phenylmethylsulfonyl fluoride.
 - e. exact composition of the cytoskeleton buffer used to prepare 4% paraformaldehyde could be given

****Referees cross-commenting****

I also mentioned in teh significance section the two weak points (only one non-cancer cell line (RPE-1); the precise role of the LINC complex). I thus think all three reviewers come to a similar conclusion: technically well done albeit some improvements are possible (reviewer 2). Manuscript is interesting but whether the findings can be generalized remains open and the overall impact is limited. Personally, I think a good strategy for the authors might be to stay with the three cell lines and avoid too general statements.

2. Significance:

Significance (Required)

General assessment: This is a very elaborate analysis of centrosome positioning at the entry of mitosis. The experiments are carefully controlled and the findings supported by multiplied experiments, e.g. the aspect of mitotic cell rounding by analysis of unperturbed cells but also by manipulation accelerating and inhibiting cell rounding. Contribution of the LINC complex is evaluated by shRNA against SUN1/2, i.e. main LINC components, but also by the KASH-DN fragment, which acts as dominant negative. On the downside the study is limited to one untransformed cell line. Given that the treatments interfering with LINC complex function most likely affect all aspects of LINC-centromere interplay, it remains open what precise function of the LINC-complex contributes to chromatin segregation fidelity

Advance: The work clearly shows that at least in RPE-1 cells the separated centrosomes arrange each other along the shorter axis of the nucleus and that the LINC complex is required for this.

Audience: The work is certainly interesting for researchers interested in mitosis, most precisely in spindle assembly. It enlightens a very specific aspect of spindle assembly but this very convincing. The work is basic research.

Our experience is basic research of mitosis, nuclear structure and function both using biochemical assays and life cell imaging.

September 29, 2023

Re: Life Science Alliance manuscript #LSA-2023-02404-T

Prof. Jorge G. Ferreira
Instituto de Investigação e Inovação em Saúde -i3S
Rua Alfredo Allen
Porto 4200-315
Portugal

Dear Dr. Ferreira,

Thank you for submitting your manuscript entitled "The LINC complex ensures accurate centrosome positioning during prophase" to Life Science Alliance. We invite you to re-submit the manuscript, revised according to your Revision Plan.

Thank you for this interesting contribution to Life Science Alliance. We are looking forward to receiving your revised manuscript.

Sincerely,

B. MANUSCRIPT ORGANIZATION AND FORMATTING:

Life Science Alliance manuscript #LSA-2023-02404-T**Point-by-point reply to the reviewers' comments and suggestions****Reviewer 1:**

- *“Moreover, we demonstrate this mechanism is altered in cancer cells, leading to increased chromosome segregation errors. » Here the authors infer that the identified mechanism is absent in cancer cells and that its absence contributes to chromosome segregation errors. Both conclusions are not supported by the presented data. First, the authors did not test whether any members of the LINC complex or dynactin is present at lower levels on the nuclear membranes of the cancer cells. Such a direct validation would be essential to make such a strong statement. Second, the authors conclude that this mechanism prevents chromosome segregation errors, based on the fact that depletion or impairment of the LINC complex (shSUN1, shSUN2, DN-KASH) results in chromosome segregation errors. These perturbations lead, however, as noted by the authors themselves to pleiotropic effects, including insufficient retraction of nuclear membrane, which can all contribute to chromosome segregation errors. It is therefore impossible to estimate the contribution of the centrosome positioning mechanism to these segregation errors using this type of perturbations. One could even argue that this mechanism might not be that important, since depletion of SUN2, which also impairs centrosome positioning has no significant effect on chromosome segregation.*

We agree with the reviewer that an analysis of the levels of LINC complex components and dynactin in cancer cells is lacking. For this reason, we have now analyzed the levels of SUN1, SUN2, dynactin and nesprins by immunofluorescence in RPE-1 as well as the cancer cell lines. This was added to a new supplementary figure S5 and is discussed in the text.

In addition, considering the reviewer's concern related to the pleiotropic effects of LINC complex depletion on chromosome segregation (with which we fully agree), we decided to remove this section and rewrite the manuscript and conclusions to focus on the role of the LINC complex in centrosome positioning only.

- *“The authors conclude based on three cell lines that the centrosome positioning mechanisms is present in non-transformed cells and not in cancerous cells. The authors have, however, only analysed 1 non-cancerous cell line, and they compare cells originating from vastly different tissues (retina, bones and breast) and origins (epithelial vs. mesenchymal cartilage cells). Such*

a general statement is not possible without a systematic comparison of several healthy cells vs cancerous cells from the same tissue”.

We agree with this reviewer’s comment, which is also shared by the other reviewers. Accordingly, we have now extensively rewritten the manuscript to tone down this statement and focus on the role of the LINC complex in determining centrosome positioning.

- “While the data showing that centrosome positioning depends on the LINC complex is solid and robust, some of the “negative” examples identified by the authors are less convincing. One the process the authors study is cell rounding. Based on the fact that Rap1 transfection or treatment with Calyculin A does not lead to differences that are statistically different, the authors conclude that cell rounding is not involved. However, absence of statistical difference does not mean that there is no difference. Indeed, when comparing the raw data in Figure 2L and 2Q to the positive hit shSun2 in Figure 4J, one could conclude that cell rounding does make a difference, and that this statistical difference would emerge if the authors would count a high number of cells. Therefore the authors should interpret these results in a more differentiated manner, and also instead of just stating nonsignificant, state also the real p-values for the different experiment”.

According to the reviewer’s suggestion, we have now added all p values to the respective graphs and interpreted these results in a more step-by-step manner. Moreover, while we understand the reviewer’s comment regarding our sample size, it should be noted that this is a single-cell, high-resolution imaging approach which, in combination with certain treatments makes it very challenging to obtain data for a high number of cells. In this regard, we point out that interfering with cell rounding was extremely difficult to achieve. When highly overexpressed, Rap1* completely impairs mitotic cell de-adhesion, and this blocks mitotic entry (Marchesi et al., 2014). Furthermore, CalA treatment induces a fast and drastic rounding, which makes it very challenging to accurately track centrosome and nuclear positions. Nevertheless, we filmed additional cells treated with CalA and added the data to the figures. Our results still confirm that interfering with cell rounding does not significantly change centrosome positioning during this stage. We would also like to note that the sample size in all conditions is within the range normally used when performing single-cell high resolution imaging.

- The second major concern emerges when looking at the data in Figure 5, when the authors test for the abundance of the dynein complex on the nuclear envelope in cells treated with DPPPL-KASH or DN-KASH. Yes, there is a statistical difference, but the absolute difference is tiny (I estimated a normalized intensity of 1.44 vs 1.35). This is a difference of less than 10%. How do the authors think that such a small change in dynein could have such a strong effect on centrosome positioning? Would a partial dynactin depletion by 10% give an equivalent result? Does the depletion of other proteins involved in the late recruitment of dynein at the NE also affect centrosome positioning?

We thank the reviewer for this important point. Originally, we quantified dynactin intensity by selecting three unbiased random regions of the NE. However, it is possible this approach underestimates the overall fluorescence intensity across the entire structure. For this reason, we have now developed a custom-designed MATLAB algorithm to measure dynactin fluorescence intensity over the entire NE using the same dataset. We have replaced Fig. 5K and L with this data and a description of the method has been added to the Materials and Methods section. As can be seen from the new graph, there is a reduction of approximately 50% in dynactin NE fluorescence intensity.

The reviewer also asks whether depletion of other proteins involved in the late recruitment of dynein at the NE would also affect centrosome positioning. However, extensive previous work done by us and others, has shown that depletion of either BicD2 or NudE/NudEL, which are the main adaptors for dynein loading during the G2/M transition, significantly affect prophase centrosome positioning, since they detach centrosomes from the NE (Splinter et al., 2010; Bolhy et al., 2011; Hu et al., 2013; Baffet et al., 2015; Nunes et al., 2020). Once detached, centrosomes are no longer able to orient according to nuclear cues (Nunes et al., 2020). Therefore, we do not believe such an approach would provide additional information regarding the role of the LINC complex in this process.

Reviewer 2

“The authors need some other NE protein as a control to show that the reduction of dynein by DN-KASH is a specific defect and not a broad impact on the NE. The dynein data in Figs. 5J-L need to be extended to SUN1/2”.

We thank the reviewer for these suggestions. To clarify this point, we have analyzed the levels of lamin B following expression of DN-KASH or □PPPL-KASH. This data was added to Fig. S4. This allowed us to conclude that expression of the DN-KASH construct affects dynein, but likely does not impact

other NE proteins. In addition, we have also analyzed dynactin levels following SUN1 and SUN2 depletion. This data have been included in Figure 5.

- *"Figure 1 is insufficiently explained. The authors have to describe in an understandable way how they measured centrosome-centrosome angle and centrosome-nucleus angle. They should show a cartoon in which these angles are clearly shown. The small cartoons in Fig. 1C are not helpful at all; they are also not explained. The authors should explain the meaning of the black dots (are these centrosomes?) and the even smaller dots. The short nuclear axis should be indicated, e.g., by a red line".*

We apologize for the lack of sufficient explanation in Figure 1. We have now rewritten the text to clarify all the points. We have also added a scheme explaining how centrosome-nucleus and centrosome-centrosome angles are quantified, according to the reviewer's suggestion. We have added this to Fig. S1. We believe this makes our data more understandable and easier to follow.

"On the first page of the manuscript: "Consequently, at the NEP, centrosomes are positioned on the shortest nuclear axis (Fig. 1C) as can be seen in Fig. 1A. This means that the centrosome-nucleus angle relative to the shortest nuclear axis should be 0. However, in Fig. 1C, this angle is between 45 and 90 degrees. This is also the case for Fig. 1D. Please clarify".

We thank the reviewer for noticing this error. In fact, the graphs should always reflect positioning of centrosomes relative to the longest nuclear axis. Therefore, when the values are close to 90°, this means they are oriented on the shortest nuclear axis. We understand this could be confusing to the readers. We have now clarified this information throughout the text.

- *"I find it confusing that in Fig. 1, depending on the subfigure, the short or longest nuclear axis is used as a reference point: Fig. 1C: shortest; D: shortest; F: shortest; G: longest; I: shortest; J: longest. Thus, even within the same cell line, the reference point is changing. What is the rationale for this variation"?*

Again, we refer to the point above. The reference point is always the shortest nuclear axis. However, we apologize for the lack of clarity. This has all been changed according to the explanation provided in the previous point.

- *Fig. 4K, L, M: in figure, y-axis: "shortest nuclear axis". In legend: "relative to the longest nuclear axis". I guess the longest nuclear axis is correct. Same in Fig. 5D and E. Fig. 5C lacks the WT control.*

This information has been clarified in the text and panels have been corrected accordingly. Regarding Fig. 5C, we believe the correct control is the expression of \square PPPL-KASH, since it has been shown extensively that nesprins localize to the NE in control, unmanipulated cells. Nevertheless, we have added a WT control to Supplementary Figure 5, showing localization of nesprins in unmanipulated prophase cells.

“The cells in Fig. 5J are not comparable: one has a monopolar spindle, the other a bipolar. The authors need some other NE protein as a control to show that the reduction of dynein by DN-KASH is a specific defect and not a broad impact on the NE. The dynein data in Figs. 5J-L need to be extended to SUN1/2”.

We agree with the reviewer’s comment that the cell in the top panel might appear as a monopolar. However, it is not. In fact, this cell has centrosomes on the top and bottom of the nucleus, in a vertical configuration, a common feature of RPE-1 cells (check Magidson et al., Cell, 2011). To clarify this, we have now added lateral projections of all cells, highlighting the centrosomes to clearly show they are positioned on opposite sides of the nucleus.

The other points related to the effects of DN-KASH on other NE proteins and dynactin levels following shSUN1 and shSUN2 were also addressed and added to the figures.

“The title of the paper is misleading: the authors do not provide any indication for a nuclear signal in prophase that determines centrosome positioning”.

We have changed the title of the manuscript according to the reviewer’s suggestion.

“It would make sense to use the same time scale in Figs. 1A and B (either min.sec. or sec.) to allow direct comparison”.

We have now changed the time scale to seconds in all figures to allow direct comparison.

“2nd section: Mitotic cell rounding “The authors state: Given that cancer cells failed... I would be careful with this generalization; only one cancer cell was used in this study”.

Given the limited number of cells that we used, and following the concern raised by all reviewers, we have now re-written the text to avoid generalizations regarding cancer cell lines. Instead, we now focus on the role of the LINC complex in determining centrosome positioning.

"The authors say: "However, they did not place the centrosomes at the shortest nuclear axis (Figure 4K-M)." Centrosomes are still on the shortest nuclear axis but not as frequent as in control".

This has been corrected.

"The white color in Fig. 6B cannot be seen and needs to be changed to something else".

We apologize for this oversight. During the upload and pdf conversion process, we did not realize the color of this bar, corresponding to the DN-KASH group had changed to white. However, given the concerns raised by reviewer 1 relative to the pleiotropic effects of SUN depletion, we have now removed the data concerning chromosome segregation and focus instead on the centrosome positioning mechanism.

The paper has neither line nor page numbers.

This has been added.

Reviewer 3:

"Fig. 3: I suggest to quantify the lamin B1 and LBR overexpression levels".

According to the reviewer's suggestion, we have quantified the levels of lamin B1 and LBR in cells overexpressing the respective constructs. These data were added to Fig. S3.

"it would make sense to indicate the test used for each p-value in all the figure legends".

We have now added the statistical test used and the p-value in the figure legends.

"Figure legends are quite repetitive and could be shortened. E.g. in Fig. 1 the description for E, F, H and I repeats what has been explained for B and C. Same

applies between figure legends. The authors might refer to previous legends if the analysis was done in a similar way”.

We have now simplified the legends.

“How is nuclear solidity defined and analyzed in Fig S3D”?

Nuclear solidity was analyzed using Fiji. In short, nuclei are outlined using the polygon tool and nuclear area is measured. To calculate nuclear solidity, the nuclear area is then divided by the corresponding nuclear convex hull area. Irregular nuclei will typically show a lower nuclear solidity value. This information was added to the manuscript.

“The references to Fig S3 in figure legend 3 (“see Fig S3”) do not enlighten the message and could be removed. The same applies to Fig5 - here it is not clear why the author refer to Fig S4”.

We agree with this reviewer’s comment. We have now removed these references from the legends.

“Fig. 5: Consider reordering the panel: Start with the current panel C (as in the text) as it is the necessary control prior to the experimental data”.

We have now changed the order of the panel according to the reviewer’s suggestion.

“Fig 5 I: what means “before”? Can the authors give a time window they use for analysis”.

We have now replaced the term “before” with a defined time.

*“Page 20: “... shortest nuclear axis (Fig. 1C, 5D-G; n.s. - not significant). However, DN-KASH-expressing cells showed compromised separation and positioning of centrosome (Fig. 5D-G, * $p=0.0155$ and * $p=0.0237$, respectively). - rather point to the specific panels, i.e. Fig. 1C, 5D and F as well as and Fig. 5E and G”.*

We have now clarified these points in the text.

"Fig 6B. The DN- KASH bars are on my pdf not visible - use a darker grey".

As mentioned above, we apologize for this oversight. We did not realize that during the pdf conversion process the bar corresponding to the DN-KASH group had changed to white. However, considering our reply to a concern raised by reviewer 1 (please see comments above), we have now decided to remove the section related to the chromosome segregation phenotypes and focus on the centrosome positioning mechanism instead.

"Fig S6, albeit mentioned in the text, is not included in the supplementary info".

We apologize for this error. In fact, where it reads Fig. S6, should be Fig. S5. We have now corrected this.

"a. GlutaMAX instead of GlutaMAXE (page 29)

b. What means "as described previously"? No reference is given. Do you refer to the upper part of the method section? (page 30)

c. 20 nM HEPES should most probably read 20 mM (page 32)

d. "1:50 protease inhibitor; 1:100 Phenylmethylsulfonul fluoride" - which protease inhibitor (mixture)? Rather phenylmethylsulfonyl fluoride.

e. exact composition of the cytoskeleton buffer used to prepare 4% paraformaldehyde could be given".

All these suggestions/corrections have been introduced in the text.

December 18, 2023

RE: Life Science Alliance Manuscript #LSA-2023-02404-TR

Prof. Jorge G. Ferreira
i3S - Instituto de Investigação e Inovação em Saúde, Universidade do Porto
Rua Alfredo Allen
Porto 4200-315
Portugal

Dear Dr. Ferreira,

Thank you for submitting your revised manuscript entitled "The LINC complex ensures accurate centrosome positioning during prophase". We would be happy to publish your paper in Life Science Alliance pending final revisions necessary to meet our formatting guidelines.

- Tables should be numbered consecutively with Arabic numerals (1, 2, 3, 4); They can be included at the bottom of the main manuscript file or be sent as separate files.
- please add your main, supplementary figure, table, and movie legends to the main manuscript text after the references section
- we encourage you to revise the figure legend for Figure 5 such that the figure panels are introduced in an alphabetical order
- please add callouts for Figures 4L,M; 5B,C,F; 6A,B to your main manuscript text

A. FINAL FILES:

B. MANUSCRIPT ORGANIZATION AND FORMATTING:

Sincerely,

Reviewer #1 (Comments to the Authors (Required)):

The authors have revised the manuscript and sufficiently addressed all points raised by the reviewers.

Reviewer #2 (Comments to the Authors (Required)):

The LINC complex ensures accurate centrosome positioning during prophase

Lima et al.

The authors have thoroughly revised the manuscript. The comments that I have raised are addressed. Importantly, they explain experimental set ups and data much better. The manuscript reached publication quality.

January 4, 2024

RE: Life Science Alliance Manuscript #LSA-2023-02404-TRR

Prof. Jorge G. Ferreira
i3S - Instituto de Investigação e Inovação em Saúde, Universidade do Porto
Rua Alfredo Allen
Porto 4200-315
Portugal

Dear Dr. Ferreira,

Thank you for submitting your Research Article entitled "The LINC complex ensures accurate centrosome positioning during prophase". It is a pleasure to let you know that your manuscript is now accepted for publication in Life Science Alliance. Congratulations on this interesting work.

DISTRIBUTION OF MATERIALS:

Again, congratulations on a very nice paper. I hope you found the review process to be constructive and are pleased with how the manuscript was handled editorially. We look forward to future exciting submissions from your lab.

Sincerely,
